EMBO
Molecular Medicine

# An allosteric inhibitor of *Mycobacterium tuberculosis* ArgJ: Implications to a novel combinatorial therapy

Archita Mishra[1] iD, Ashalatha S Mamidi[1], Raju S Rajmani[2], Ananya Ray[1], Rajanya Roy[1] & Avadhesha Surolia[1,*] iD

## Abstract

The existing treatment regime against tuberculosis is not adequate, and novel therapeutic interventions are required to target *Mycobacterium tuberculosis (Mtb)* pathogenesis. We report Pranlukast (PRK) as a novel allosteric inhibitor of *Mtb*'s arginine biosynthetic enzyme, Ornithine acetyltransferase (*Mt*ArgJ). PRK treatment remarkably abates the survival of free as well as macrophage-internalized *Mtb*, and shows enhanced efficacy in combination with standard-of-care drugs. Notably, PRK also reduces the 5-lipoxygenase (5-LO) signaling in the infected macrophages, thereby surmounting an enhanced response against intracellular pathogen. Further, treatment with PRK alone or with rifampicin leads to significant decrease in *Mtb* burden and tubercular granulomas in *Mtb*-infected mice lungs. Taken together, this study demonstrates a novel allosteric inhibitor of *Mt*ArgJ, which acts as a dual-edged sword, by targeting the intracellular bacteria as well as the bacterial pro-survival signaling in the host. PRK is highly effective against *in vitro* and *in vivo* survival of *Mtb* and being an FDA-approved drug, it shows a potential for development of advanced combinatorial therapy against tuberculosis.

**Keywords** combinatorial therapy; drug repurposing; infectious disease; *Mycobacterium tuberculosis*; Ornithine acetyltransferase
**Subject Categories** Microbiology, Virology & Host Pathogen Interaction; Pharmacology & Drug Discovery

## Introduction

Tuberculosis (TB) accounts for 1.5 million deaths worldwide every year, with a high percentage of individuals primarily from developing nations (WHO, 2012). Emergence of drug-resistant strains has further increased the *Mycobacterium tuberculosis* (*Mtb*)-associated lethality. In last decade, itself there have been enormous attempts toward the development of novel therapeutic approaches to combat multi-drug-resistant (MDR) and extensively drug-resistant (XDR) *Mtb* strains (Makarov *et al*, 2009, 2014; Koul *et al*, 2011). Novel therapy regimens endorse strategies wherein the pre-approved drugs for other ailments could be re-purposed for targeting lethal Mtb strains (Zumla *et al*, 2013). The major challenges in finding a suitable target for anti-*Mtb* drug discovery is its ever-evolving stride and the conserved nature of the essential proteins (Zuniga *et al*, 2015). Although the past decade has seen major developments in the TB drug discovery pipeline, still there is a constant need for improved therapeutic interventions (Zumla *et al*, 2013).

Traditionally, the replication machinery has been at the heart of drug discovery and the processes associated with logarithmic growth phase are exploited for drug targeting. However, targeting these vital cellular components may cause serious non-specific effects to the host. On the other hand, the intricate network of metabolic pathways provide novel avenues for specific targeting of pathogens (Boshoff *et al*, 2004; Tran *et al*, 2017). Among these, arginine biosynthesis pathway is essential for the survival and pathogenesis of *Mtb*. Despite the acknowledged significance of arginine biosynthesis in *Mtb*, inhibitors to target this pathway remain to be discovered. The enzymes involved in this pathway are promising targets for anti-TB drug development (Gordhan, 2002; Mdluli & Spigelman, 2006). Moreover, inhibitors of this pathway may provide novel insights to the significance of arginine biosynthesis in *Mtb*-associated stress responses and persistence.

One of the crucial enzyme from this pathway, Ornithine acetyltransferase (ArgJ) from *Mtb*, has been implicated as essential gene for the survival and virulence of the pathogen (Sassetti & Rubin, 2003; Sassetti *et al*, 2003). *Mt*ArgJ catalyzes the transfer of acetyl moiety from N-acetyl Ornithine to glutamate, thereby producing Ornithine and N-acetyl glutamate for next round of arginine biosynthesis (Xu *et al*, 2007; Appendix Fig S1A). Significantly, the absence of a homologous protein in human genome makes *Mt*ArgJ an exciting target for drug development. *Mycobacterium tuberculosis* argJ gene is encoded by the ORF Rv1653 and belongs to the N-terminal

1 Molecular Biophysics Unit, Indian Institute of Science, Bangalore, India
2 Microbiology and Cell Biology, CIDR, Indian Institute of Science, Bangalore, India
 *Corresponding author. Tel: +80 2293 2714; E-mail: surolia@mbu.iisc.ernet.in

nucleophile fold family of enzymes (Cole *et al*, 1998; Xu *et al*, 2007). The crystal structure of *Mt*ArgJ in native form and in complex with Ornithine has been determined at 1.7 and 2.4 Å, respectively (Sankaranarayanan *et al*, 2010). ArgJ in *Mtb* is a mono-functional enzyme as it facilitates the transfer of acetyl group to glutamate exclusively from N-acetyl Ornithine. However, ArgJ is reported to be bi-functional in some bacterial species like *Neisseria gonorrhoeae*, *B. subtilis*, *Geobacillus stearothermophilus*, and *T. neapolitana*, wherein it can utilize both N-acetyl Ornithine and acetyl-CoA for the transfer of acetyl moiety (Xu *et al*, 2007).

In the present study, we identified a previously unknown allosteric site on *Mt*ArgJ and report the discovery of a novel inhibitor that binds and impedes the catalytic efficiency of *Mt*ArgJ. The selectivity and specificity of this inhibitor lies in its ability to allosterically modulate the substrate-binding interface. Through a series of *in silico*, biochemical and biological approaches, we conclude the potency of this compound as a drug candidate against *Mtb* survival and pathogenesis (Fig 1A). Our data demonstrate the exquisite potency of this inhibitor against arginine biosynthesis in *Mtb*, thereby abating the pathogen survival, in both *in vitro* and *in vivo* infection models. We show that it also targets the pathogen pro-survival pathways in the host, thereby causing an enhanced reduction in the intracellular *Mtb* survival. Significant effect of this inhibitor in combination with the standard-of-care therapeutic regimen attests to its promise for inclusion in our armamentarium against tuberculosis. Taken together, this study identifies a novel metabolic inhibitor of *Mtb,* and its potential for improved combinatorial therapy against tuberculosis.

# Results

### Cloning, expression, purification, and characterization of MtArgJ

To target the arginine biosynthesis in *Mtb*, we selected Ornithine acetyltransferase (*Mt*OAT/*Mt*ArgJ), a crucial enzyme involved in the arginine biosynthesis pathway. *Mt*ArgJ catalyzes the transfer of acetyl moiety from N-acetyl Ornithine to glutamate, thereby producing Ornithine for arginine production (Appendix Fig S1A). *Mt*ArgJ belongs to the N-terminal nucleophile (Ntn) fold class of enzymes, synthesized as a 404-amino acid long protein, which undergoes an auto-proteolysis event between the Ala199 and Thr200. This auto-proteolysis generates two fragments of approximately equal size (20–21 kDa), which then associate to form a protomeric unit (AB—heterodimer; A2B2 tetramer—dimer of the heterodimer; Sankaranarayanan *et al*, 2010). The affinity-purified His-tagged *Mt*ArgJ thus yields three distinct bands on SDS–PAGE, one full-length and two cleaved products (20 and 21 kDa each; Fig 1B; Marc *et al*, 2001). Since we got a fraction of uncleaved (inactive) protein also during purification, for calculating the concentration of active *Mt*ArgJ in solution, we used densitometric analysis of SDS–PAGE profile (details in Appendix Materials and Methods section). The enzymatic activity of *Mt*ArgJ thus obtained was ascertained by a TLC-based assay wherein the enzyme was found to be functionally active (Appendix Fig S1B). The protein was assayed at varying substrate concentration, that is, N-acetyl Ornithine, and the reaction velocity was plotted against the corresponding substrate concentrations. The $K_m$ for N-acetyl Ornithine-mediated synthesis of Ornithine was

determined to be 91.8 μM (Fig 1C and D). These results confirm the integrity and functional activity of *Mt*ArgJ employed in this study.

### Identification of a potential ligand binding pocket on *Mt*ArgJ surface

We started our quest to discover a small molecule inhibitor of *Mt*ArgJ by investigating the protein surface. *Mt*ArgJ structure (PDB ID: 3IT6) was probed for surface cavity predictions (by MetaPocket server and CastP analysis) and a well-defined pocket of area 2,019.7 Å² and volume 3,104.8 Å³ located in-between the two active sites was discovered (Fig 2A). This pocket comprises of 48% hydrophobic amino acid residues and 24 and 28% polar and charged residues, respectively (Fig 2A inset, Appendix Fig S1D and E). This large pocket comprised of four loops, four helices, and one β strand contributed by each protomer (AB) of A2B2 tetramer. However, we noticed two flanking loops positioned at the interface of the substrate-binding pockets on either side. To experimentally validate the hydrophobicity of this pocket, we used a fluorescent dye, 8-anilinonaphthalene sulfonate (ANS). It has specificity for the hydrophobic regions of a protein and shows a characteristic fluorescence at 470 nm upon binding (Cardamone & Puri, 1992). We utilized this property of ANS to probe the novel pocket on *Mt*ArgJ. As shown in Fig 2B, ANS showed a concentration-dependent increase in binding to MtArgJ, manifested as a steady rise in fluorescence intensity. The kinetic analysis determined a $K_d$ of 937 μM (ANS binding to *Mt*ArgJ; Fig 2C). Also, in our blind docking experiment with whole molecule of *Mt*ArgJ, 2,000 final conformations were generated and we observed that all the conformers of ANS were sitting and interacting exclusively with this novel pocket (data not shown). In concordance with MetaPocket analysis, these results further validate the presence of a major cavity on *Mt*ArgJ surface, which is considerably hydrophobic in nature. We proceeded with characterization of this pocket and as shown in Fig 2D, addition of ANS leads to a decrease in the catalytic activity of *Mt*ArgJ. However, the effect was only marginal, which could be attributed to the relatively smaller size of ANS with respect to the binding pocket. Based on our initial results, we hypothesized that a suitable ligand bound to this pocket may cause inhibition of the enzymatic activity of the protein. Moreover, targeting this novel pocket instead of "substrate-binding site" was deemed by us to be more rewarding, since substrate for *Mt*ArgJ are small molecules involved in multiple cellular process. Therefore, conventional substrate analogs (as inhibitors) may cross-react with other cellular proteins with similar ligands, giving rise to unwanted side effects. Our initial results gave us a lead to further probe this pocket for inhibitor development against *Mt*ArgJ.

### *In silico* screen of FDA-approved drug library against *Mt*ArgJ

We performed a high-throughput *in silico* screen of a small molecule drug library to determine their binding to the major pocket on *Mt*ArgJ. Importantly, chosen library was FDA-approved therefore already cleared for cytotoxicity tests and poised for therapeutic repurposing. From the dataset of 1,556 FDA-approved drugs, a total of 1,417 compounds were prepared and sourced into the virtual screening pipeline along with the reference molecule ANS. Initially flexible docking filtered 1,340 compounds based on their internal

**A**

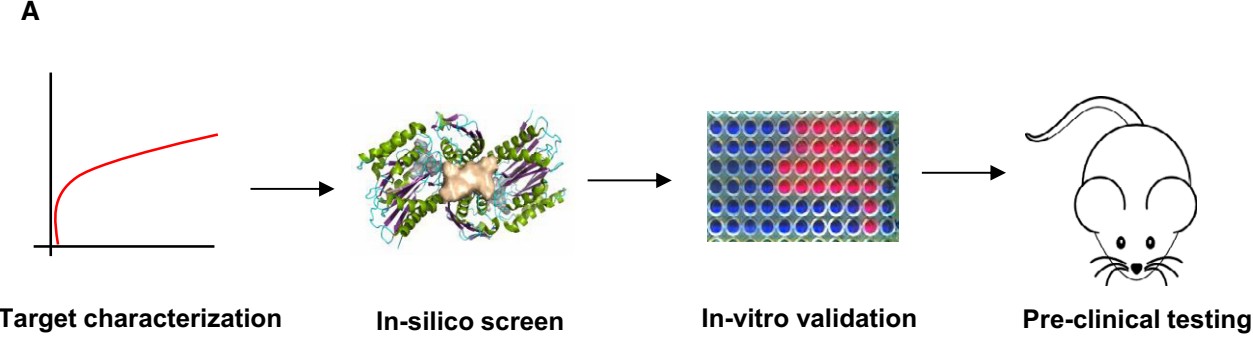

**Target characterization**    **In-silico screen**    **In-vitro validation**    **Pre-clinical testing**

**B**

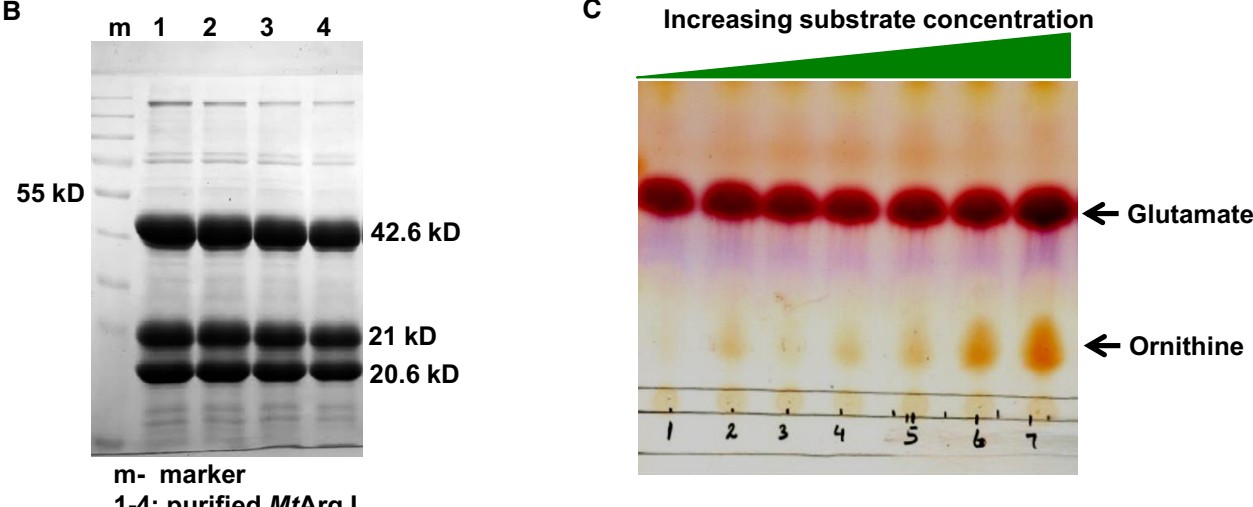

**C**

**D**

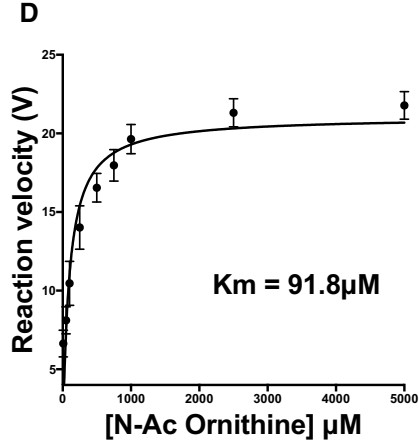

**Figure 1.  Screening methodology and characterization of *Mt*ArgJ target enzyme.**

A    Screening methodology employed in the study.
B    SDS–PAGE profile of *Mt*ArgJ purified by His-tag affinity chromatography.
C    *Mt*ArgJ activity assay where product (Ornithine) formation is monitored by a TLC-based assay, and TLC image is shown with increasing substrate (N-acetyl Ornithine) concentration; lower spots represent the product Ornithine, and upper spots represent the other substrate glutamate (kept constant).
D    Saturation curve fit using Michaelis–Menten plot for *Mt*ArgJ activity as measured by TLC assay. Reaction velocity on *y*-axis represents amount of product formed per unit time, while *x*-axis denotes substrate concentrations. Mean and standard error (SE) determined from three experimental replicates ($n = 3$) and two technical duplicates.

                    

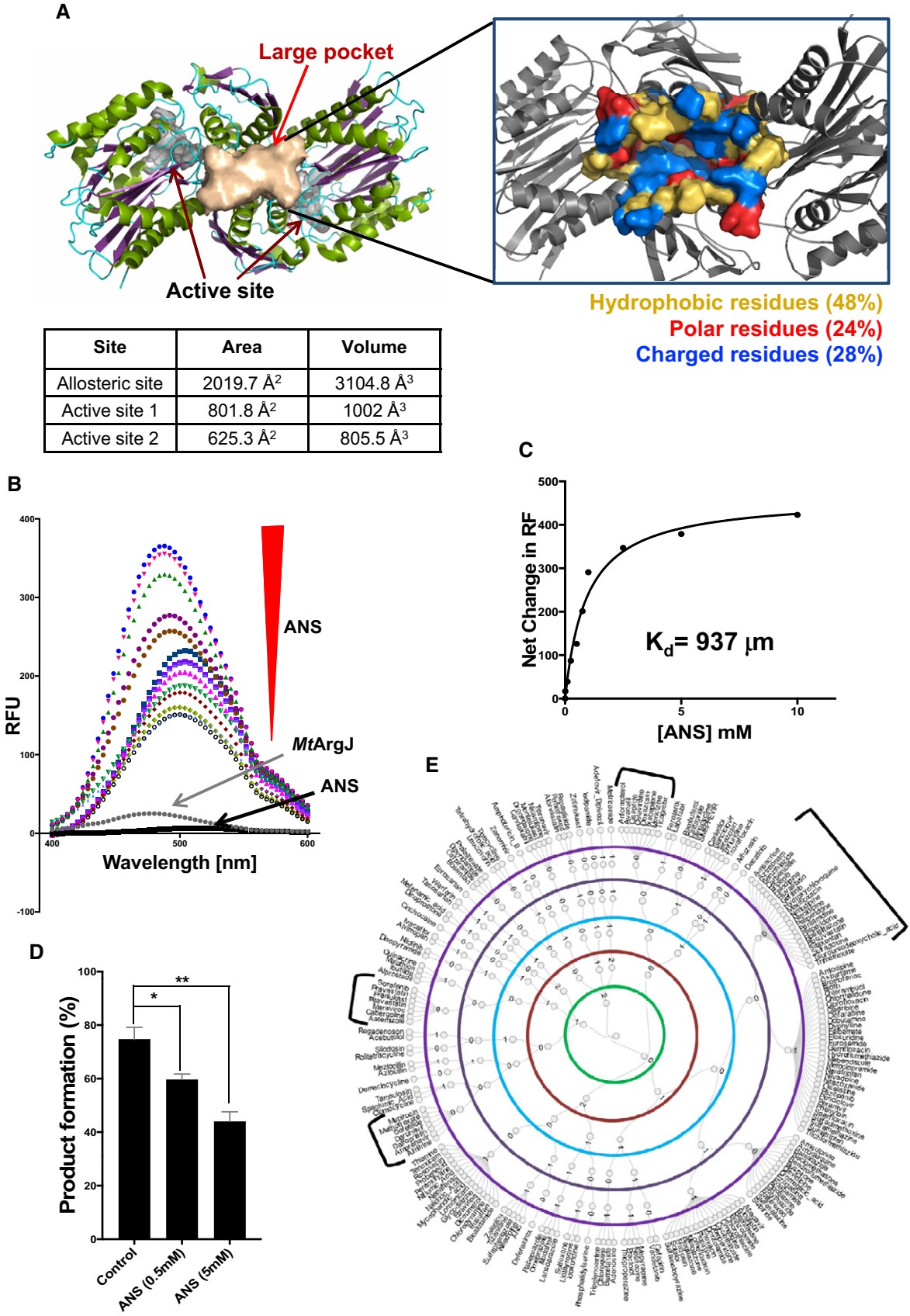

**Figure 2.**

◄

**Figure 2.  Characterization of allosteric site on *Mt*ArgJ surface and its role in enzyme activity.**

A  Cartoon representation of *Mt*ArgJ with the large pocket (allosteric site) represented in wheat color, with one active site on either side. Inset shows the composition of hydrophobic, polar, and charged amino acid residues in the pocket (PDB ID 3IT6).

B  Fluorescence-based assay: The graph shows increase in fluorescence upon ANS binding to *Mt*ArgJ, in a concentration-dependent manner (ANS = 10 μM–10 mM).

C  Dose-dependent net change in relative fluorescence of ANS upon *Mt*ArgJ binding plotted against ANS concentration for $K_d$ determination.

D  Decrease in *Mt*ArgJ activity in the presence of ANS as determined by TLC-based assay. Mean and standard error (SE) determined from three experimental replicates ($n = 3$) and two technical duplicates. Statistical significance between experimental groups was determined by two-tailed, unpaired Student's *t*-test  (\*\**P* < 0.01, \**P* < 0.1).

E  In silico screen for allosteric modulators: The hyperbolic graph (circular dendrogram) showing hierarchical clustering of selected compounds based on the five parameters represented as green (violation of Lipinski's rule of five), maroon (free binding energies), cyan (ligand strain energy), black (percentage change in total solvent-accessible surface area), and purple (gap index) determining the surface complementarity in receptor–ligand complexes.

degrees of freedom. Subsequently, grid scoring based on electrostatic energies (0.59 to −34.39 kcal/mol) and van der Waals energies (−5.43 to −109.25 kcal/mol) was applied. This bivariate histogram-based partitioning selected 743 compounds, which showed better binding to *Mt*ArgJ than ANS (Appendix Fig S2A). In the second layer of screening, Amber-based rescoring obtained 203 compounds that outperformed ANS (Appendix Fig S2B). Next, screening based on five parameters (Appendix Table S1), namely (i) violation of Lipinski's rule of five (Lipinski *et al*, 1997), (ii) binding free energies, (iii) ligand strain, (iv) change in solvent-accessible surface area (ΔSASA), and (v) gap index, segregated these 204 ligands (203 + ANS) into various clusters (Fig 2E). From these clusters, we selected four major clusters, A (0000), B (10000), C (01000), and D (11000), comprising of 43 molecules, with best possible set of scores for all five parameters mentioned above (Appendix Table S2). Hydrogen bond interactions for these 43 receptor–ligand complexes were computed and tabulated (Appendix Fig S2D and Table S2) for the assessment of their binding strength to *Mt*ArgJ. The results obtained from *in silico* high-throughput screen paved the way for secondary validation screens.

### *In vitro* validation of *in silico* predictions

To validate the virtual screening approach developed here, compounds from each subclass were selected for experimental testing (Appendix Table S3). We performed an extensive enzyme kinetic study with selected compounds by evaluating their potential to inhibit *Mt*ArgJ activity. The activity-based screen led to the discovery of two potent compounds, Pranlukast (PRK) and Sorafenib (SRB; Fig 3A and B). To determine the mechanism of inhibition, we assessed their effect on $K_m$ (Michaelis constant) and $V_{max}$ (maximum velocity) for the MtArgJ activity assay. The kinetic parameters were determined by varying substrate (NAO) concentration at multiple inhibitor (PRK/SRB) concentration during enzyme catalysis. In the absence of inhibitor, the $V_{max}$ was calculated to be 22.4 μg/min. Increasing inhibitor (PRK/SRB) concentration lowered the $V_{max}$ of the reaction without affecting the apparent $K_m$ (Fig 3C and D). These results demonstrate a non-competitive mode of inhibition of *Mt*ArgJ, orchestrated by PRK and SRB. Next, we performed Dixon plot analysis of 1/V versus inhibitor concentration (PRK/SRB) at three different substrate concentrations (0.1, 0.5, and 1 mM NAO). The data revealed a $K_i$ value of 139 μM for PRK-mediated *Mt*ArgJ inhibition while that of SRB was calculated to be 244 μM (Fig 3E and F). These results establish a non-competitive mode of *Mt*ArgJ inhibition by PRK and SRB. Further, the data indicate PRK to be an efficient inhibitor of *Mt*ArgJ activity than SRB. The results are consistent with our rationale of probing the allosteric site for

inhibition of *Mt*ArgJ. We have also performed a negative validation of our *in silico* screening strategy by testing 10 compounds from the non-selected group (filtered out), and none of them could inhibit the *Mt*ArgJ activity *in vitro* (Appendix Table S4).

### PRK and SRB bind to a novel allosteric pocket discovered on the surface of *Mt*ArgJ

To experimentally determine the binding site of PRK and SRB, we designed an ANS-based fluorescence titration assay (Iyer *et al*, 2016). The *Mt*ArgJ, saturated with ANS, gives a characteristically high fluorescence intensity at 470 nm, and any molecule that competes for ANS binding site should result in a dose-dependent decrease in fluorescence. As shown in Fig 3G and H, addition of PRK/SRB leads to diminution in fluorescence intensity at 470 nm in a dose-dependent manner. The data indicate binding of PRK and SRB to the allosteric pocket by competitive displacement of ANS from the *Mt*ArgJ complex. Both PRK and SRB were spectroscopically inert in this region. Further, net change in the relative fluorescence unit (RFU) was plotted as a function of ligand concentration to calculate binding constants. The dissociation constant ($K_i$) for PRK-induced displacement of ANS from *Mt*ArgJ was calculated to be 115 μM, whereas that of SRB was 312 μM (Fig 3G and H: inset). The $K_d$ values thus obtained for both the compounds are about 10 times lesser than that for ANS, which establishes their significant binding to the region. These results demonstrate the suggestive affinity and specificity of PRK and SRB for the allosteric pocket discovered here, at the surface of *Mt*ArgJ.

### PRK and SRB impart thermal stability to *Mt*ArgJ in a concentration-dependent manner

To further establish the binding affinity of PRK and SRB to *Mt*ArgJ, we used thermal shift assay (TSA), a method orthologous to isothermal titration calorimetry (ITC; Niesen *et al*, 2007; Iyer *et al*, 2016) and is being productively used for drug discovery (Renaud *et al*, 2016). *Mt*ArgJ with varying concentration of PRK/SRB was subjected to gradually increasing temperature, and the shift in melting temperature ($T_m$) was calculated. The extent of change in $T_m$ is indicative of the ligand's affinity for protein. As shown in Fig 3I and J, thermal stability of *Mt*ArgJ demonstrates positive correlation with increasing concentration of both the inhibitors (PRK and SRB). However, the increase in $T_m$ was relatively higher in case of PRK, consistent with its higher affinity for the protein. The apparent dissociation constant ($K_d$) for PRK and SRB, calculated by plotting net change in $T_m$ versus inhibitor concentration, was 126 and 281 μM, respectively (Fig 3I and J: inset). These results are

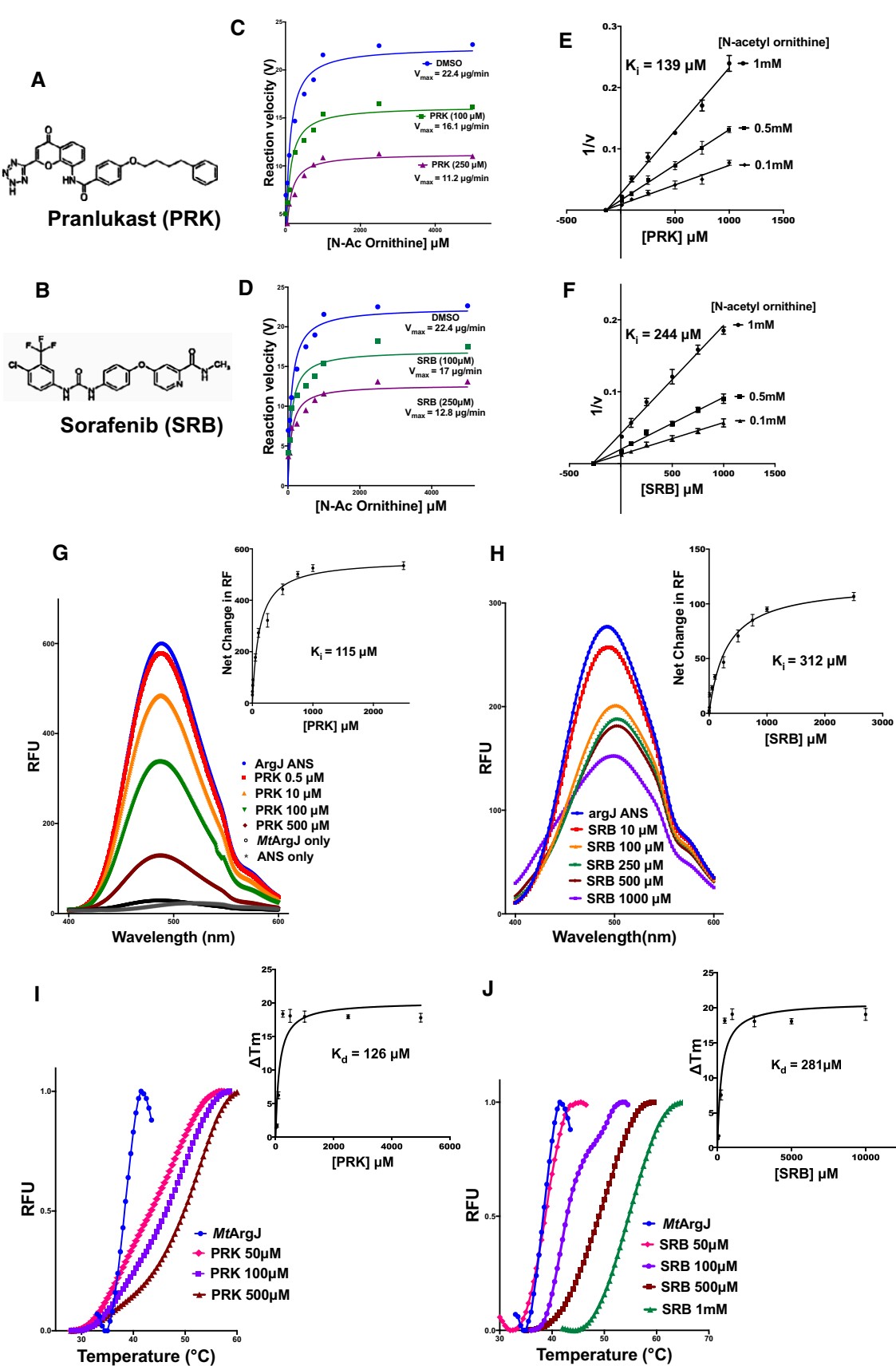

**Figure 3.**

**Figure 3.  *In vitro* kinetics and binding studies of inhibitory compounds (PRK and SRB) *Mt*ArgJ.**

A, B    Chemical structures of (A) Pranlukast (PRK) and (B) Sorafenib (SRB).
C, D    Saturation curve fit using Michaelis–Menten plot for *Mt*ArgJ activity at two different inhibitor concentrations for (C) PRK and (D) SRB, quantified by TLC-based assay.
E, F    Dixon plot analysis performed by plotting 1/v against varying inhibitor concentration, at three different substrate concentrations for (E) PRK and (F) SRB.
G, H    Fluorescence-based dye (ANS) displacement assay to determine the binding of (G) PRK and (H) SRB into the hydrophobic pocket on *Mt*ArgJ. $K_d$ values for both were determined by plotting net change in relative fluorescence upon inhibitor binding as a function of inhibitor concentration.
I, J    Thermal shift assay (TSA) to determine apparent $K_d$ for inhibitor binding to *Mt*ArgJ: with increasing concentration of (I) PRK and (J) SRB to examine the shift in melting temperature ($T_m$) of *Mt*ArgJ. Apparent $K_d$ values calculated by plotting net change in melting temperature upon inhibitor binding as a function of inhibitor concentration.

Data information:  (E–J) Mean and standard error (SE) determined from three experimental replicates ($n = 3$) and two technical duplicates.

consistent with PRK to have higher affinity for *Mt*ArgJ than SRB, thereby imparting enhanced thermal stability to the protein. Importantly, $K_d$ values determined by TSA agreed with the enzyme kinetics and fluorescence spectroscopy data (Fig 3C–H). These results validated the significant affinity of both the inhibitors for *Mt*ArgJ. However, PRK induces a more positive shift in thermal stability as compared to SRB at same concentrations. Based on the data so far, it was evident that PRK is a better inhibitor of *Mt*ArgJ and has higher affinity for the protein, than SRB. We next asked for the potential residues involved in this allosteric mode of inhibition.

### MD simulation results decipher a proposed mode of PRK/SRB binding to the allosteric pocket on *Mt*ArgJ

Based on the promising results obtained through our biochemical analysis, we sought to determine the molecular basis of PRK-mediated inhibition of *Mt*ArgJ through computational approaches. We undertook molecular dynamic (MD) simulation to examine the possible mode of PRK/SRB-mediated allosteric inhibition of *Mt*ArgJ. Hydrogen bonds contribute to the directionality and stabilization of protein–ligand complexes. Hence, the occurrence of hydrogen bonds between substrate-bound *Mt*ArgJ and PRK/SRB was examined (Appendix Tables S5 and S6). While nestled in the allosteric pocket of *Mt*ArgJ, the tetrazole ring of PRK interacts with Asp234 and Ser310, respectively, from chain B and chain D while chromene ring showed interactions with Ser310 of chain B and benzamide group nitrogen with Gln305 of chain B on the protein (Fig 4A and B). Sorafenib, on the other hand, showed interactions via amino groups to Gln305 and Arg308 of chain D and carbonyl group to Arg308 of chain B (Fig 4C and D). PRK exhibited more number of interactions with the allosteric pocket than SRB, asserting higher affinity of PRK for *Mt*ArgJ. The data show that PRK and SRB both bind to the allosteric pocket on *Mt*ArgJ; however, PRK binds with higher affinity than SRB (Fig 3C–J). Details of MD simulation results are represented in Appendix Figs S3–S5 and Appendix Results and Discussion.

To validate the results obtained from MD simulations, we performed site-directed mutagenesis of the allosteric pocket. The potential inhibitor-interacting residues identified by the MD simulation data, viz. Gln305, Ser310 and Asp234, were mutated into alanine one by one. Also, we made double mutants of Gln305/Ser310 and Asp234/Gln305. All these mutants exhibit drastic decrease in the enzymatic activity of *Mt*ArgJ, the double mutants being totally inactive (Appendix Fig S6B). These results attest to the allosteric nature of this pocket and support the results obtained from biochemical assays performed earlier (Fig 3A–J). Hence, we hypothesized that both PRK and SRB could be a potential drug candidate

for targeting *Mt*ArgJ activity in *Mtb*. This prompted us to test the dose-dependent effect of PRK and SRB on the survival of pathogenic *Mtb*.

### PRK and SRB significantly compromises the *in vitro* survival of pathogenic *Mtb*

So far, we characterized the affinity parameters and the possible mechanism involved in the PRK/SRB-based inhibition of *Mt*ArgJ. We then proceed to determine the efficacy of these inhibitors on pathogenic strain of *Mtb* (H37Rv) and the two multiple drug-resistant (MDR) strains (Jal 2261 and Jal 2287). H37Rv cells were exposed to varying concentrations of PRK and SRB. The microplate Alamar blue assay (MABA) was employed to determine the $MIC_{90}$ (minimum inhibitory concentration- 90% inhibition in cell survival) of the inhibitors. Alamar blue (AB) is an oxidation–reduction indicator dye that has been widely used to measure the sensitivity of mycobacteria to anti-TB drugs (Franzblau, 2000). A color transition from non-fluorescent blue to fluorescent pink indicating reduction in AB dye occurs during mycobacterial growth. Inhibitor-mediated depletion in growth interferes with AB reduction and subsequent color development. Administering pathogenic *Mtb* (H37Rv) with increasing concentration of PRK or SRB resulted in decreased fluorescence intensity (Fig 5A). Rifampicin (Rif) was taken as a positive control. The minimum inhibitory concentration ($MIC_{90}$) was calculated by plotting cell viability (%) against inhibitor concentration. Based on the MABA assay, the calculated $MIC_{90}$ for PRK and SRB against *Mtb* H37Rv are 5 and 10 μg/ml, respectively (Fig 5B and C). Next, the Hill's plot analysis of fluorescence intensity versus inhibitor concentration revealed the $IC_{50}$ of PRK- and SRB-mediated inhibition of *Mtb* survival to be 3.02 and 6.7 μg/ml, respectively (Fig 5D and E). The results suggest the potential anti-tubercular properties of lead compounds. However, it also indicated the superiority of PRK over SRB in inhibiting the growth and survival of *Mtb*. We also tested the efficacy of PRK and SRB on two MDR strains: Jal2261 and Jal2287. PRK showed an $MIC_{90}$ of 15 and 25 μg/ml for both Jal2261 and Jal2287 strains, respectively (Appendix Fig S7A and B). However, SRB demonstrated comparatively higher MIC (Appendix Fig S7C and D) for both the strains. The results showed the promising effect of PRK on pathogenic *Mtb* including MDR strains.

### PRK and SRB showed marked reduction in *Mtb* survival in combination with standard-of-care anti-TB drugs

After establishing the efficacy of PRK and SRB as potent inhibitor of mycobacterial growth, we tested their efficacy in combination with the standard-of-care anti-TB drugs [rifampicin (R), isoniazid (H),

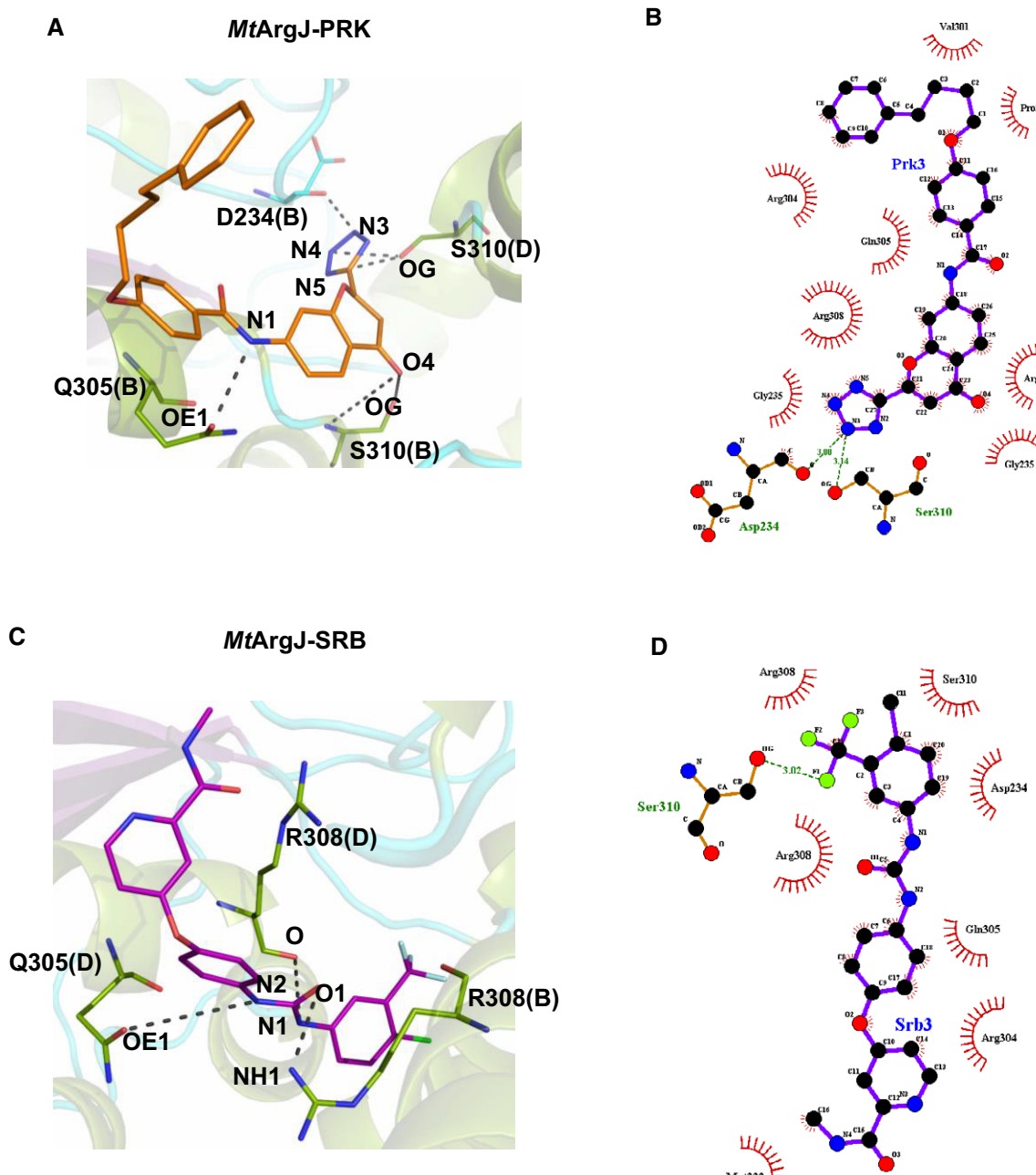

**Figure 4.  Binding interactions of PRK and SRB with the allosteric pocket of *Mt*ArgJ.**

A    PRK (orange) interacting with the Asp234 and Gln305 of B chain, Ser310 of B and D chains in the allosteric pocket of the *Mt*ArgJ.

B    LIGPLOT for the interaction of PRK with the allosteric pocket of *Mt*ArgJ.

C    SRB (magenta) interacting with Gln305 of D chain and Arg308 of B and D chains in the allosteric pocket of the *Mt*ArgJ.

D    LIGPLOT for the interaction of SRB with the allosteric pocket of *Mt*ArgJ.

and ethambutol (E)]. *Mtb* H37Rv cells were seeded in a 96-well plate for 48 h at 37°C and then treated with combination of inhibitors. The inhibitory properties of RHE combination was compared with novel combination of RH + PRK, wherein ethambutol was replaced with PRK. This enabled us to compare the efficacy of our inhibitor against a standard-of-care metabolic drug of *Mtb*, that is, ethambutol. A combination of rifampicin (40 ng/ml), isoniazid (30 ng/ml), and ethambutol (1.5 μg/ml) was used as a reference control to compare the efficacy of novel combination of rifampicin and isoniazid with PRK/SRB. The results indicate a 10-fold decrease in CFU upon RH + PRK treatment at the end of 24 h (Fig 5F). On the other hand, RH + SRB showed only a modest decrease of 0.2 log unit CFU (Fig 5G). These results demonstrate that PRK effectively inhibits *Mtb* survival and works very efficiently in combination with

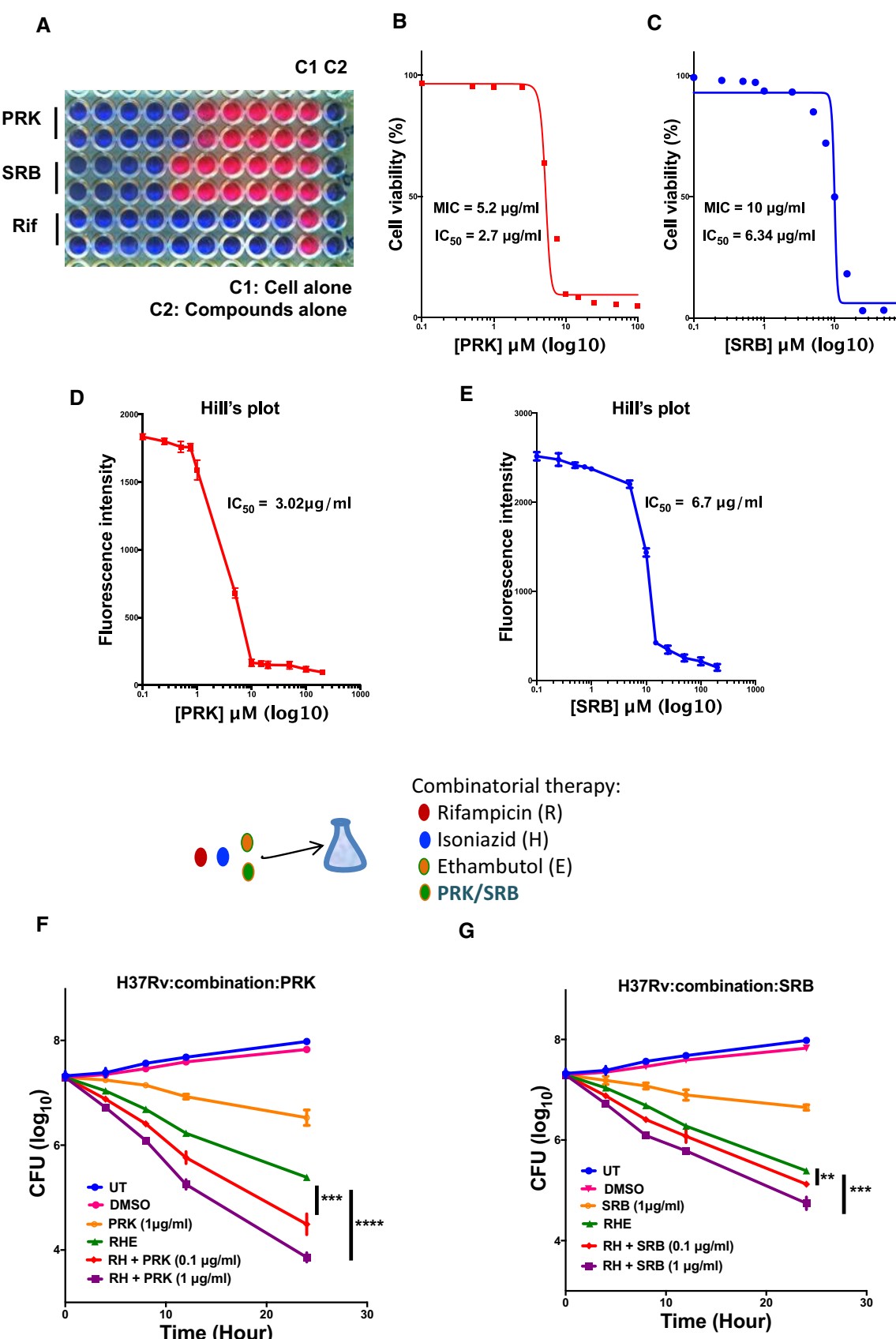

**Figure 5.**

**Figure 5.  Effect of PRK and SRB on *Mtb* H37Rv survival.**

A        Representative image of Alamar blue assay employed for MIC determination.

B, C    *Mtb* H37Rv cells were treated with varying concentration of inhibitors, and cell viability was determined using Alamar blue assay. $MIC_{90}$ was calculated by plotting cell viability (%) against increasing concentration of inhibitors PRK and SRB.

D, E    Hill's plot analysis of decline in Alamar blue fluorescence with increasing concentration of PRK and SRB, for $IC_{50}$ determination.

F, G    CFU analysis of *Mtb* treated with PRK/SRB alone or a cocktail of Rif, Inh and Emb abbreviated as RHE and compared with a new combination of RH + PRK and RH + SRB, at two concentrations of PRK and SRB (0.1 and 1 μM), respectively. The *x*-axis represents the time points post-treatment with the inhibitors, and all the experiments were performed in triplicate and confirmed with biological duplicates at least. Statistical significance between experimental groups was determined by two-tailed, unpaired Student's *t*-test (****$P < 0.0001$, ***$P < 0.001$, **$P < 0.01$, *$P < 0.1$).

Data information: (D–G). Mean and standard error (SE) determined from three technical replicates ($n = 3$) and at least two biological duplicates.

the standard-of-care anti-TB drugs (rifampicin and isoniazid). This new combination of Rif, Inh and PRK is significantly effective against *Mtb* survival than the pre-existing RHE combination and holds a potential for improved therapy regimen. Therefore, we further tested the efficacy of PRK on macrophage infection model of *Mtb*.

**PRK significantly inhibits the survival of *Mtb* in macrophage infection model without affecting the host cell**

THP-1 cells, a human monocytic cell line, stimulated for differentiation by PMA treatment were infected with pathogenic *Mtb* (H37Rv). The *Mtb*-infected THP-1 cells were treated with varying inhibitor (PRK or SRB) concentrations or DMSO (control) at different time points. Once treated, the cells were lysed at desired time points (0, 12, 24, and 48 h) and plated for colony formation assays. The CFU (colony-forming unit) is calculated and plotted against time at three different inhibitor concentrations. As shown in Fig 6A, treatment of *Mtb*-infected THP-1 cells with PRK (5 μg/ml) leads to 100-fold reduction in CFU within 48 h. Moreover, identical assay with mouse macrophages (Raw264.7) treated with PRK leads to about 200-fold decrease in CFU (Fig 6B). In comparison, SRB was much less potent and shown about 10- to 30-fold decrease in CFU for human and mouse macrophage cell lines, respectively (Fig 6A and B). These results demonstrate the superiority of PRK over SRB in reducing the mycobacterial burden from the infected macrophages.

Next, we investigated the efficacy of PRK and SRB in combination with the standard-of-care anti-TB drugs. The *Mtb*-infected macrophages were treated with either pre-existing RHE combination or novel R/H/PRK and R/H/SRB cocktails. Interestingly, treatment with RH + PRK combination exhibited almost 40- to 50-fold decrease in CFU from that of parent combination (RHE) in both human and murine macrophage cell lines (Fig 6C and D). However,

RH + SRB leads to only 0.2–0.3 log unit decrease in CFU signifying its reduced efficiency toward combination therapy (Fig 6C and D). These results demonstrate the efficacy of PRK in reducing the *Mtb* burden from host macrophages and its enhanced efficiency in combination with standard-of-care drugs.

We further demonstrate the active effect of PRK treatment on macrophage-internalized *Mtb* and its possible side effects on the host cell survival. Macrophages were infected with GFP-tagged *Mtb* H37Rv (*Mtb*-GFP henceforth) followed by treatment with PRK at varying time points. Cells were harvested and flow-sorted based on GFP expression followed by assessment of macrophage viability by PI (propidium iodide) staining. The PRK treatment leads to diminished GFP intensity with time, suggesting reduced *Mtb* burden in infected macrophages (Fig 6E and F). However, host cell (macrophage) viability remains unaffected as determined by PI staining (Fig 6G). Detailed scatter plots for the FACS experiments are shown in Appendix Fig S8A and B. These results demonstrate the potency of Pranlukast (PRK) as a promising anti-tubercular molecule with no deleterious effect on the host cell survival. Also, the enhanced effect of PRK alone and in combination with the therapy drugs, on the macrophage-internalized *Mtb*, is interesting and beneficial from the host's perspective.

**PRK treatment reduces the infection-associated apoptosis in the host**

It has been reported that during the early phase of *Mtb* infection, macrophages undergo apoptosis as an innate defense mechanism, thereby increasing the levels of pro-apoptotic proteins like caspases 3 and 8 (Duan *et al*, 2002; Derrick & Morris, 2007; Behar *et al*, 2011; Aguiló *et al*, 2014). Hence, we explored the effect of PRK on infection-associated apoptosis in the host. To examine this, we choose to monitor caspase 3-dependent apoptosis in *Mtb*-infected

**Figure 6.  Effect of PRK and SRB on the macrophage-internalized *Mtb*.**

A, B    THP1 (human monocytic cell line) cells and Raw264.7 (mouse macrophage cell line) cells were infected with *Mtb* H37Rv followed by treatment with PRK (5 and 25 μM) and SRB (10 μM). CFU of internalized *Mtb* plotted at defined time points.

C, D    CFU analysis of THP1- and Raw264.7-internalized *Mtb* upon treatment with a cocktail of Rif, Inh, and Emb abbreviated as RHE and compared with a new combination of RH + PRK (0.5 μM) and RH + SRB (0.5 μM).

E, F    Flow cytometry analysis of macrophage-internalized *Mtb* H37Rv-GFP at 4, 8, 16, and 32 h post-treatment with PRK (5 μM).

G       Flow cytometry analysis of *Mtb* (H37Rv-GFP)-infected macrophages stained with propidium iodide (PI) dye to determine macrophage cell death upon *Mtb* infection and PRK treatment.

H, I    Effect of PRK treatment on infection-induced apoptosis in macrophages by monitoring active caspase 3 levels in supernatant media of *Mtb*-infected THP1 and Raw264.7 macrophages in presence and absence of PRK (1 and 10 μM).

J       Relative decrease in chemiluminescence as a measure of extracellular caspase 3 levels upon PRK treatment.

Data information: All the experiments were performed in triplicate and confirmed with biological duplicates at least. Statistical significance between experimental groups was determined by two-tailed, unpaired Student's *t*-test (****$P < 0.0001$, ***$P < 0.001$, **$P < 0.01$). Mean and standard error (SE) determined from three technical replicates ($n = 3$) and at least two biological duplicates.

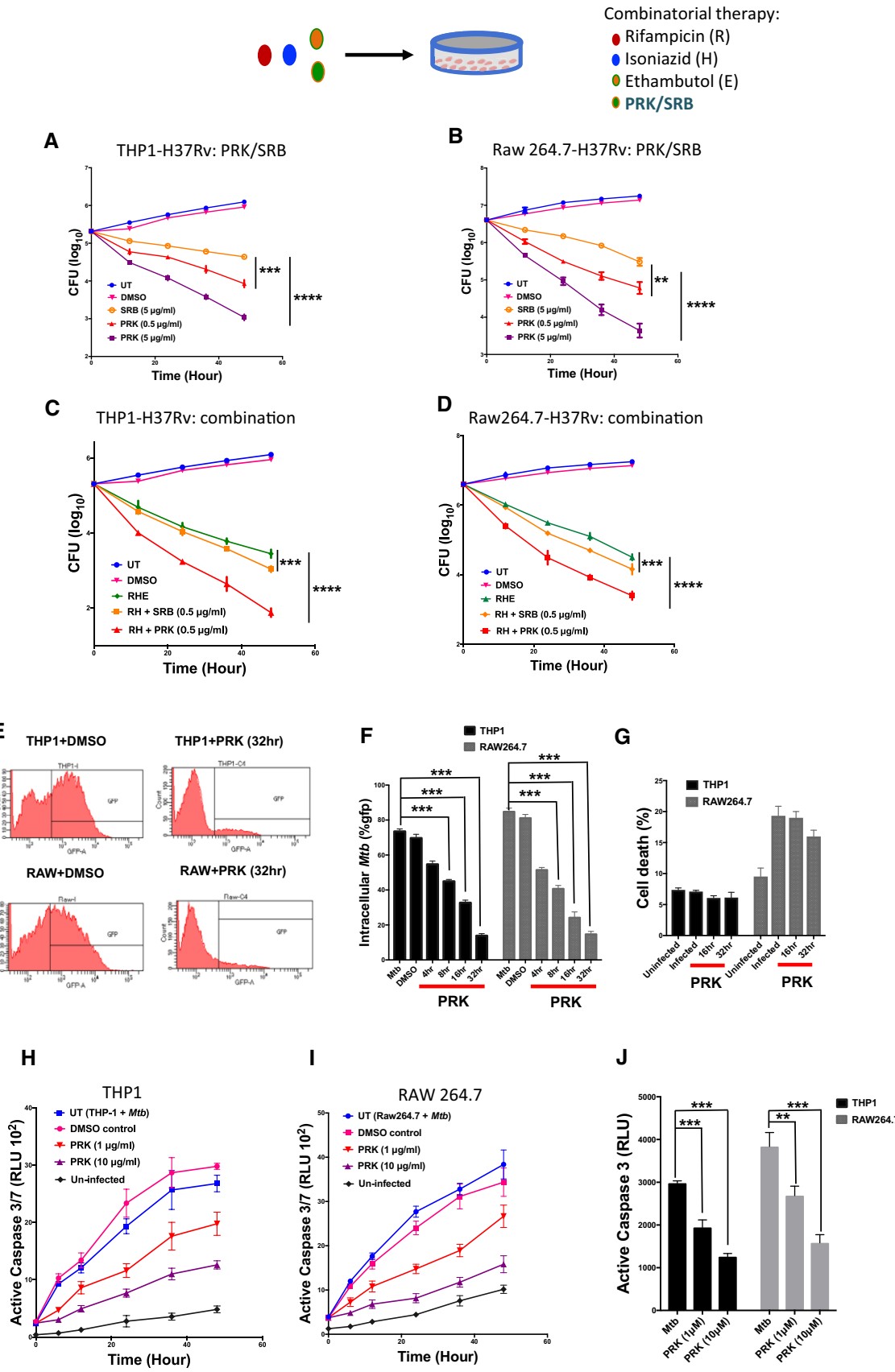

**Figure 6.**

macrophages. As shown in Fig 6H–J, treatment with increasing concentrations of PRK leads to reduced extracellular caspase 3 level in the *Mtb*-infected human as well as murine macrophages (THP-1 and Raw 264.7, respectively). However, uninfected macrophages showed no increase in caspase 3 levels while the infected macrophages showed a gradual increase in caspase 3 with time (up to 48 h). The results demonstrate decrease in *Mtb*-associated macrophage apoptosis upon PRK treatment in both human and mouse macrophage cell lines. The above results suggest that PRK not only targets the macrophage-associated *Mtb* but also limits the infection-induced host cell apoptosis. These results instigated us to further test the *in vivo* efficacy of PRK in mouse models of infection.

### PRK-mediated killing of *Mtb* is rescued upon arginine supplementation

Our *in vitro* studies showed that PRK efficiently inhibits the activity of *Mt*ArgJ, a crucial enzyme in the arginine biosynthesis pathway of the pathogen. So, we hypothesized that the *in vivo* (cell-based) effect is due to the reduction in arginine levels within the pathogen. Therefore, we aimed to confirm the mechanism of action of PRK-mediated killing of *Mtb* H37Rv. To examine this, we supplemented the *Mtb* minimal media cultures with arginine alongside PRK treatment and calculated the CFU post-treatment. As shown in Fig 7A, the *Mtb* cultures treated with PRK (1 µg/ml) had significant decrease in the cell survival. However, upon supplementation with arginine (1 mM), the effect of PRK-mediated cell death is rescued. In the samples treated with rifampicin (0.4 µg/ml), arginine supplementation had no effect on the cell death. This shows that PRK induces its bactericidal effect on *Mtb* by reducing the arginine levels in the pathogen, while no such effect was observed when a totally different antibiotic (the one against bacterial RNA pol) was used instead. The inability of *Mtb* to survive under arginine deficit created by PRK is also consistent with the suitability of *Mt*ArgJ as a target against *Mtb* survival.

We then sought to confirm the decline in arginine levels in PRK-treated samples through mass spectrometry. We treated the Mtb cultures with PRK or DMSO, and the whole-cell metabolites were isolated and subjected to ESI-MS analysis. The area and intensity of the arginine peak ($m/z$ 174) was determined, and the arginine levels were estimated based on a standard arginine plot. A heavy isotope-labeled arginine (13C labeled arginine: $m/z$ = 180) was used as an internal standard (Appendix Fig S9A and B). As shown in the Fig 7B, the levels of arginine in the *Mtb* H37Rv (wild type) cells are calculated to be 1.7 µg/ml. However, in the PRK-treated samples,

there was a significant decline in arginine levels, as low as 150 ng/ml, at higher inhibitor concentration (0.5–5 µg/ml). We got similar results by estimating the levels through either peak area or peak intensity (Appendix Fig S9C). These results demonstrate that PRK treatment actively reduces the arginine levels in *Mtb* cells, thereby attenuating the cell growth and survival of the pathogen.

### PRK also targets the host macrophage leukotriene signaling to limit the intracellular *Mtb* growth

The above results showed that PRK is potent in reducing the *Mtb* growth by inhibiting the pathogen's arginine biosynthesis pathway and is highly effective on the macrophage models of infection as well. However, the enhanced effect of PRK on the macrophage-internalized *Mtb* suggests for an additional target within macrophages, aiding to potent reduction of pathogen survival within the host. PRK is a known inhibitor of cysteinyl leukotriene receptor-1 (CysLTR1), on the mammalian cells and is used for treatment of asthma (Barnes & Pujet, 1997). Macrophages also express CysLTR1 in response to various inflammatory stimuli, including pathogenic bacterial colonization. Also, studies have shown that in macrophages and dendritic cells, PRK acts through a yet another mechanism, wherein it targets the leukotriene and prostaglandin (eicosanoids) biosynthesis, which are ligands for CysLTRs (Theron *et al*, 2014). Phospholipase-A2 converts membrane phospholipids to arachidonic acid, which is used as a precursor to synthesize leukotrienes and prostaglandins by the enzymes like 5-LO (5-lipoxygenase), FLAP (5-lipoxygenase-activating protein), and COX-2 (cyclooxygenase 2; Drazen, 1998; Mayer-Barber *et al*, 2014). Divangahi *et al* have shown that *Mtb* infection activates the 5-lipoxygenase pathway, which facilitates the host cell necrosis, thereby helping the pathogen dissemination. This also prevents the cross-antigen presentation by dendritic cells, thereby inhibiting the induction of T-cell immunity (Behar *et al*, 2010; Divangahi *et al*, 2010).

We have shown here that CysLTR1 and 5-lipoxygenase (5-LO) genes are upregulated by 4.5-fold and 3.5-fold, respectively, in the macrophages infected with *Mtb* (Fig 7C). Along with that, FLAP and COX-2 were also significantly upregulated by *Mtb* infection (Fig 7C). Notably, upon PRK treatment, we observed a remarkable decline in the transcript levels of 5-LO (4.5-fold), Cox-2 (8.5-fold), and FLAP (2.5-fold; Fig 7D–F). This shows that PRK downregulates the 5-lipoxygenase pathway, thereby reducing the *Mtb* survival and dissemination within the macrophages. Also, PRK treatment causes a notable decline in the CysLTR1 transcript levels in the macrophages, along with downregulation in the MCP-1 levels, a

**Figure 7. PRK targets the arginine biosynthesis in *Mtb* and 5-lipoxygenase signaling in the *Mtb*-infected host macrophages.**

A   CFU analysis of *Mtb* H37Rv treated with DMSO control, PRK, PRK with arginine supplement, Rif, and Rif with arginine supplement. As observed, arginine supplementation could reverse the effect of PRK on *Mtb*. As a control, arginine supplementation had no effect on the rifampicin-mediated cell death.

B   ESI/MS analysis of the whole-cell metabolites isolated from PRK-treated *Mtb* samples. DMSO-treated samples were taken as control. Concentration of arginine in each sample was calculated by a standard arginine plot (inset).

C   qPCR analysis of the genes CysLTR1, 5-LO, FLAP, COX-2, and MCP1 in the *Mtb*-infected versus uninfected macrophages (Raw 264.7).

D–H  qPCR analysis of the genes (D) 5-LO, (E) COX-2, (F) FLAP, (G) CysLTR1, and (H) MCP1 upon PRK treatment in the infected macrophages, DMSO and Rif treatment as controls. As observed, 5-lipoxygenase and associated genes involved in eicosanoid biosynthesis were significantly downregulated upon PRK treatment but not by Rif treatment.

Data information: All the experiments were performed in triplicate and confirmed with biological duplicates at least. Statistical significance between experimental groups was determined by two-tailed, unpaired Student's *t*-test (\*\*\*$P < 0.001$, \*\*$P < 0.01$). Mean and standard error (SE) determined from three technical replicates ($n = 3$) and at least two biological duplicates.

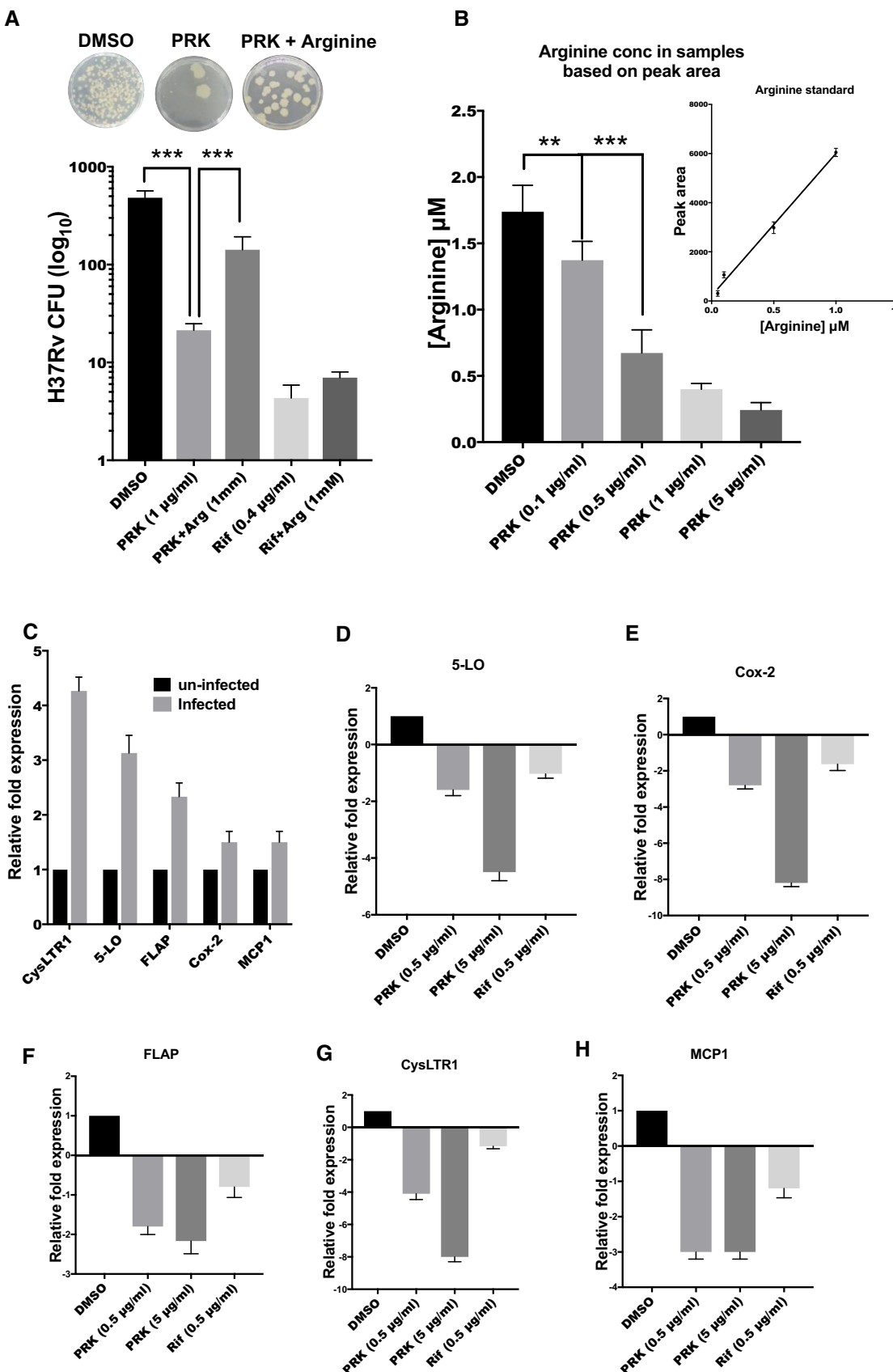

**Figure 7.**

downstream effector of CysLTR1 signaling (Ichiyama *et al*, 2005, 2009) (Fig 7G and H). The decrease in the CysLTR1 gene expression could be due to two reasons: (i) since the leukotriene biosynthesis is inhibited, CysLTR1 is downregulated in the absence of its ligand (leukotrienes), (ii) since CysLTR1 levels are upregulated upon *Mtb* infection, PRK-mediated reduction in *Mtb* within macrophages will lead to reduction in the CysLTR1 levels as well. Taken together, these results demonstrate the enhanced effect of PRK on the macrophage-infected *Mtb*. In addition to targeting the arginine biosynthesis in *Mtb*, PRK also targets the host 5-lipoxygenase signaling, thereby potentiating its bactericidal effect in the macrophage model of infection. This study also highlights the importance of 5-LO and leukotriene signaling in the *Mtb* pathogenesis.

### PRK treatment reduces the *Mtb* burden and tubercular granulomas from the lungs of *Mtb*-infected mice

Chronic *Mtb* infection is characterized by the formation of lung-associated granulomas, an organized aggregate of immune cells with infected macrophages at the core (Ramakrishnan, 2012). To investigate the effect of PRK on chronic *Mtb* exposure, we used BALB/c mice and infected them with *Mtb* H37Rv strain through aerosol (Fig 8A). One month after successful establishment of infection, we treated the mice with PRK (40 mg/kg body weight), Rif (10 mg/kg body weight), or combination of PRK with Rif. Next, these mice were sacrificed at three time points (0, 12, and 24 days) and lung-associated granulomas were analyzed (Fig 8B–E). We observed a marked reduction in the tubercular granulomas after 12 and 24 days of treatment with PRK in comparison with PBS (phosphate-buffered saline)-treated mice (Fig 8B and C). Notably, Rif + PRK combination had practically no visible granulomas in the mice lungs, after 24 days of treatment (Fig 8D and E). To determine the lung-associated bacterial burden, we homogenized the mice lungs and plated at different dilutions on 7H11 solid media supplemented with OADC and PANTA. The plates were incubated for 21 days, and colonies were analyzed for CFU count. We observed significant reduction in *Mtb* burden, with a 0.5 log unit decrease in CFU, in PRK-treated mice as compared to PBS control (Fig 8F). Notably, PRK in combination with standard-of-care anti-TB drug, Rif, showed improved results with maximum diminution of lung-associated *Mtb* burden and a decrease in CFU by 1 log unit as compared to treatment with Rif alone (Fig 8G). The number of granulomas per tissue section of the mice lungs was calculated by the H&E staining analysis of the lung slides (Fig 8H and I). We observed

that there was a significant decrease in the tubercular granulomas in the PRK-treated mice. However, PRK showed most remarkable effect in combination with rifampicin. Moreover, there was no splenic or hepatic cytotoxicity at the administered dosages, as shown in the detailed histopathological H&E staining images, analyzed by expert pathologists (Appendix Figs S10 and S11). The results demonstrate the potency of PRK in combating *Mtb* infection and its improved efficiency in combination with Rif, thereby proving its *in vivo* efficacy on the pre-clinical model of tuberculosis.

## Discussion

The importance of arginine biosynthesis pathway in the survival and pathogenesis of *Mycobacterium tuberculosis* is well established. However, there are no attempts to target these metabolic enzymes for anti-TB drug discovery. One of the key enzyme from this pathway is Ornithine acetyltransferase (OAT/*Mt*ArgJ) that recycles the acetyl group during the process of arginine biosynthesis. Targeting *Mt*ArgJ for anti-TB drug development is advantageous for two main reasons: (i) It is an essential gene for the survival and virulence of the pathogen (Sassetti & Rubin, 2003; Sassetti *et al*, 2003), and (ii) it lacks a homologous protein in human (host) genome. The later imparts specificity and subsequently minimizes the potential of cross-reactivity. Although there are reported inhibitors for various metabolic pathways of *Mtb*, inhibitors that target arginine biosynthesis, more specifically the enzyme Ornithine acetyltransferase (*Mt*ArgJ) remains unexplored (Neres *et al*, 2012; Capodagli *et al*, 2014; Palde *et al*, 2016). The substrates for arginine biosynthesis are small molecules common to many other cellular pathways, including those in the host. Hence, designing a substrate analog as an inhibitor is often accompanied with severe off-target effects in such systems of host–pathogen interaction. On the other hand, allosteric sites are evolutionarily less conserved and hence provide selectivity and specificity for drug targeting with minimum side effects (Wenthur *et al*, 2014). Therefore, we rationalized that an inhibitor targeting allosteric site may unlock the way for inhibiting arginine biosynthesis pathway and thus the survival of *Mtb*, while minimizing the side effects.

In the present study, we employed structure-based *in silico* and functional *in vitro* strategies to characterize mycobacterial ArgJ for drug targeting. We first characterized a novel allosteric pocket on *Mt*ArgJ and established its relevance in modulating enzyme activity. The fluorescence spectroscopic studies validated the partial

**Figure 8. PRK treatment reduces the lung-associated granuloma from *Mtb*-infected mice.**

A    Lung images of mice infected with chronic *Mtb* infection through aerosol and treated with PRK, intraperitoneally, and/or Rif, orally, for 24 days.

B–E  (B) Lungs of mice treated with PBS and (C) PRK, after 12 and 24 days, respectively. The white spot (cyst like) depicted with an arrow corresponds to tubercular granulomas (D) Lungs of mice treated with Rif alone and (E) Rif + PRK after 12 and 24 days of infection.

F    CFU analysis of *Mtb* from lungs of infected mice treated with PBS or PRK at 0, 15, and 24 days of treatment.

G    CFU analysis of *Mtb* from lungs of infected mice treated with Rif and Rif + PRK at 0, 15, and 24 days of treatment. Appropriate negative and positive controls taken as PBS- and Rif-treated mice, respectively (*n* = 6, six mice per condition, for all the time points and dosages and **$P < 0.01$ is significant).

H    Histopathology-based granuloma analysis (blind) of mice lung tissues was done, and the number of granulomas per tissue section was plotted for PBS-treated, PRK-, Rif-, and Rif + PRK-treated mice.

I    Histopathology-based H&E staining of the lung tissue sections of mice treated with PBS versus PRK, Rif, and Rif + PRK treatments (representative images, detailed images in Appendix Figs S10 and S11). L, lymphocytes; FC, foamy macrophages; PMNs, polymorphonuclear cells.

Data information: Statistical significance between experimental groups was determined by two-tailed, unpaired Student's *t*-test (***$P < 0.001$, **$P < 0.01$). Mean and standard error (SE) determined from six biological replicates (*n* = 6).

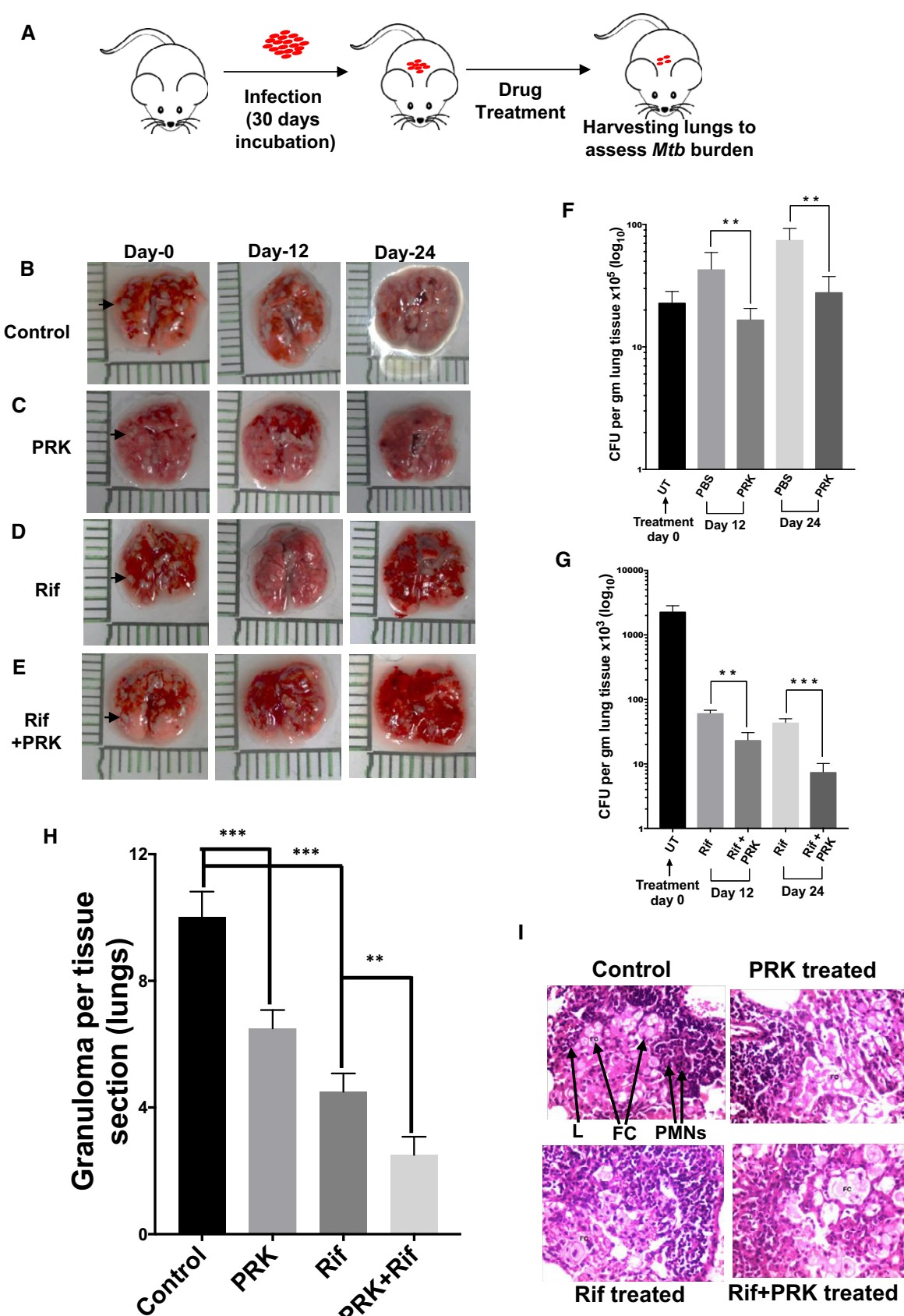

**Figure 8.**

hydrophobic nature of this pocket and its potential as an inhibitor-binding site. The results lead to the identification of two lead compounds, Pranlukast (PRK) and Sorafenib (SRB) from the library of 1,556 (FDA-approved) drugs. Pranlukast (PRK) is an FDA-approved small molecule antagonist of human cysteinyl leukotriene receptor-1 (hCysLTR1) and is recommended for the treatment of chronic bronchial asthma (Barnes & Pujet, 1997), Whereas Sorafenib (SRB) is an imatinib derivative, used for the inhibition of tyrosine kinases implicated in advanced renal-cell carcinomas (Escudier *et al*, 2007). In our study, we extensively characterized both the inhibitors (PRK and SRB) for their biochemical and biological properties. We showed that both the compounds bind to the novel pocket on *Mt*ArgJ, thereby allosterically modulating the substrate-binding and subsequent enzymatic activity. Our activity-based assay demonstrates PRK and SRB to inhibit *Mt*ArgJ activity in a noncompetitive manner. PRK showed higher affinity for *Mt*ArgJ in both fluorescence-based and thermal shift assays and was efficiently inhibiting the catalysis at lower concentrations than SRB.

The MD simulation analysis of *Mt*ArgJ-NAO complex with PRK/SRB suggested plausible interactions involved between the inhibitor and the allosteric site. PRK showed four critical hydrogen bond interactions with the allosteric pocket and lead to destabilization of major catalytic residues (viz. Thr200) involved in *Mt*ArgJ activity.

Similar investigation of *Mt*ArgJ_NAO complex bound to SRB showed lesser interactions than PRK. It is important to note that allosteric inhibition is not very common and this report is first of a kind for any Ornithine acetyltransferases (OAT) to our knowledge. Here, we highlight a conceptual advance of harnessing a site topologically distinct from the catalytic one, as an Achilles heel for compromising a target essential to the survival of a pathogen.

Further, treatment of *Mtb* H37Rv with PRK or SRB showed a marked reduction in mycobacterial survival. It also reduced the mycobacterial burden from infected human and murine macrophage models. In concordance with biochemical data, PRK was more potent than SRB against both extracellular and intracellular *Mtb* survival. We demonstrate that PRK, while killing the intracellular *Mtb*, does not have a detrimental effect on the host cell survival. Although, *Mtb* induces anti-apoptotic signals during the latent phase of infection, there are various reports showing *Mtb*-infected macrophages to undergo apoptosis as their early defense mechanism against infection, thereby increasing the levels of pro-apoptotic proteins like caspase 1, 3, 5, 7, and 8 (Duan *et al*, 2002; Derrick & Morris, 2007; Behar *et al*, 2011; Aguiló *et al*, 2014). Our results show that PRK reduces the Mtb-induced apoptosis in macrophages, thereby rescuing them from infection-associated cell death at early time points (up to 48 h post-infection). Notably, PRK shows an

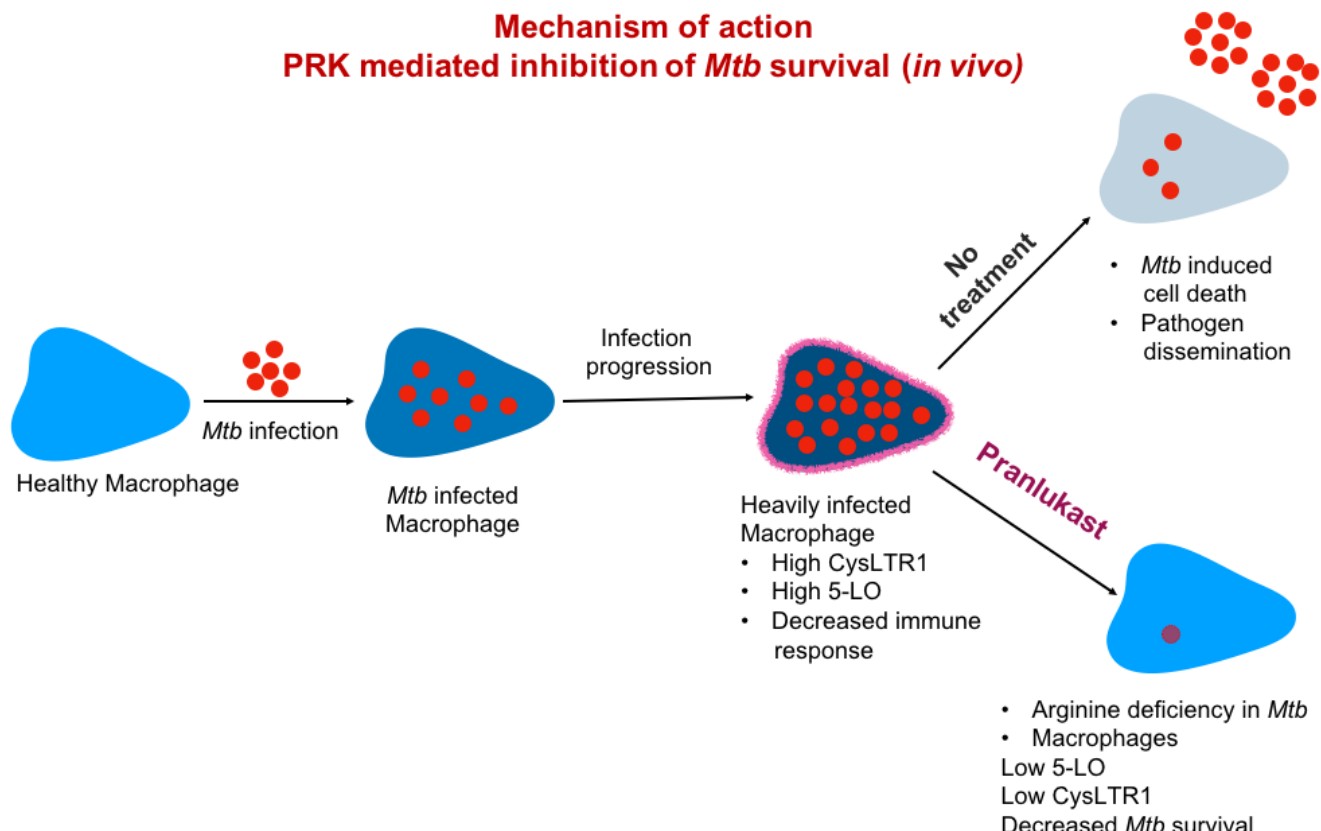

**Figure 9.  Pranlukast (PRK): A dual-edge sword for *Mycobacterium tuberculosis*.**
A representative schematic for the mechanism of PRK-mediated reduction in *Mtb* survival and pathogenesis. We show that PRK inhibits the essential arginine biosynthesis in *Mtb*, which leads to bacterial cell death. Simultaneously, PRK treatment also reduces the pathogen-specific pro-survival pathways (5-LO signaling, eicosanoid biosynthesis, and CysLTR1 signaling) in the macrophage infected with *Mtb*, thereby enabling efficient intracellular elimination of pathogen.

additive effect in combination with the standard-of-care anti-TB drugs. In combination with rifampicin and isoniazid, PRK showed significant reduction in *Mtb* survival and performed better than a pre-existing drug—ethambutol. Further, we demonstrated that PRK treatment leads to an active decline in the arginine levels in pathogenic *Mtb*. Supplementation of the *Mtb* culture with arginine rescues the PRK-mediated killing of the pathogen. This proves that the mechanism of action of PRK-mediated killing of *Mtb* is via inhibition of arginine biosynthesis.

Eicosanoid (leukotrienes, prostaglandins, etc.) signaling has been long associated with the bacterial infections and inflammation, also in the case of *Mtb*. Separate studies have shown that CysLTR1 inhibitor Pranlukast also inhibits the lipoxygenase signaling in the macrophages and dendritic cells. In this study, we show that PRK acts as a dual-edge sword, wherein it targets arginine biosynthesis in the pathogen and 5-lipoxygenase (5-LO) signaling in the host, which is known to facilitate *Mtb* survival in the macrophages. By reducing the expression of eicosanoid biosynthetic enzymes (5-LO, FLAP, and COX-2), PRK treatment effectively clears out *Mtb* burden from the infected macrophages. This way, PRKs exhibit an enhanced effect on the macrophage-internalized *Mtb*, by directly targeting the intracellular bacteria as well as enabling the host cell to combat the pathogenic attack, as represented in the schematic Fig 9.

Tubercular granulomas are a major hallmark of the successful infection and dissemination of *Mtb* in the infected lungs (Ramakrishnan, 2012). PRK showed a marked decrease in the tubercular granulomas in the lungs of *Mtb*-infected mice models, when administered alone or in combination with Rif. We observed significant reduction in *Mtb* burden from the lungs of infected mice upon PRK treatment, as determined by CFU analysis of infected lung tissues. Also, we observed notable decline in the lung-associated tubercular granulomas in the infected mice, upon PRK treatment, both alone and in combination with rifampicin. These results thus highlight the enormous potential of direct repurposing of PRK toward novel anti-TB therapeutics.

Our results unveil the potential of an *ab-initio* approach for repurposing pre-approved drugs for novel ailments. Since PRK is being administered as drug against chronic bronchial asthma, its safety toward human is already approved. Therefore, it has the potential for subjection into direct clinical trials on tuberculosis patients in near future. The compelling advantage of PRK in combination with standard-of-care anti-TB drugs and its efficacy in pre-clinical model of tuberculosis upholds its promising as an anti-TB drug. A continued effort toward clinical translation of PRK is required for development of a novel and efficient combinatorial therapy against tuberculosis.

# Materials and Methods

### *In vitro* biochemical assay for *Mt*ArgJ activity

The dialyzed *Mt*ArgJ (100 nM) was incubated with different concentrations of N-acetyl Ornithine (10 nM to 10 mM) and glutamate (0.5 mM) in 10 mM Tris–NaCl buffer (pH 7.4) at 37°C for 45 min. The reaction mixture was then loaded on a silica-TLC plate. The chromatographic run was performed in the solvent A (isopropanol:formic acid:water; 20:1:5). 100 μl reactions were made and 10 μl of it was loaded on TLC. The TLC plate was treated with 1% ninhydrin solution in methanol mixed with 5% citric acid solution. The plate is heat-dried, and the spots corresponding to Ornithine (identified by Ornithine standard) were densitometrically quantified by Multi Gauge V3.0 software, using Ornithine standard curve. All the experiments were done in triplicate, and the average was plotted using GraphPad Prism-7 software.

### Fluorescence-based dye displacement assay

Fluorescence spectroscopic studies were performed on Jasco FP-6300 Spectrofluorometer. ANS (8-anilinonaphthalene-1-sulfonic acid; Sigma) was used as a dye to probe the hydrophobic region on *Mt*ArgJ. The excitation wavelength for ANS was 374 nM while the emission was recorded from 400 to 600 nM. The excitation and emission slit widths were 2.5 and 3 nM, respectively. Each spectrum was an average of six consecutive scans, and all the experiments were done in duplicates. Protein sample (1 μM) was diluted in PBS (pH 7.4) and was checked for the "inner filter effect" over the range of ANS as well as ligand concentrations. Blank containing equal concentration of fluorophore as that in samples was used as control, and the necessary corrections were done accordingly (Cardamone & Puri, 1992). ANS concentrations were determined spectroscopically using an extinction coefficient of $6.8 \times 10^3$ $M^{-1}$ $cm^{-1}$ at 370 nM. The protein was first titrated against varying concentrations of ANS to determine the kinetic parameters of ANS binding to *Mt*ArgJ. As the ANS forms a complex with protein, a blue shift was observed (from 520 to 470 nM). 5 mM of ANS was used with 1 μM of protein to attain maximum saturation at 4°C, and the total concentration of ANS and protein was kept constant throughout the experiment. Further, the ANS-*Mt*ArgJ complex was titrated against increasing concentrations of PRK/SRB in separate experiments. The decrease in fluorescence due to the displacement of ANS by PRK/SRB from hydrophobic pocket on the protein was monitored (Iyer *et al*, 2016). To determine the dissociation constant, net change in the relative fluorescence unit (RFU) was plotted against increasing PRK/SRB concentration (at constant protein and ANS concentrations). GraphPad Prism-7 was used for the analysis, and the non-linear regression (curve fit) model (least-squares ordinary fit) with the binding saturation "one site-specific binding" function was applied.

### Fluorescence-based thermal shift assay (TSA)

A fluorescence microplate reader (iQ5, Bio-Rad iCycler Multicolor Real-Time PCR detection system) was used to monitor protein unfolding as a function of temperature. The detection involves increase in fluorescence of a fluorophore SYPRO Orange (Sigma, S5692) upon binding to the hydrophobic regions of the gradually unfolded protein. Protein samples were mixed with Hepes buffer (10 mM, pH 7.4) consisting of 1× SYPRO orange dye and appropriate concentrations of ligand over the range of several folds. Samples were incubated in 96-well PCR microplates (Bio-Rad) in the iCycler iQ5 Multicolor RT–PCR detection system. *Mt*ArgJ was kept constant at 10 μM, and the PRK/SRB concentrations were varied from 100 nM to 5 mM. DMSO was kept constant at 2% throughout the screens. According to the experimental protocol, samples were heated at 0.5°C per minute, ranging from 10 to 95°C, and the

fluorescence intensity was measured at the interval of 0.5°C (Niesen *et al*, 2007; Iyer *et al*, 2016). Cy5 filter with red-orange color intensity was selected for the SYPRO Orange detection. All the experiments were done in triplicate, and the average of three was plotted in the graphs. Appropriate buffer blanks were kept, and the intensities were subtracted for the same in each set. The fluorescence intensities were plotted as a function of temperature using GraphPad Prism-7 software, and the melting temperature of protein was determined using sigmoidal dose response function of GraphPad. The net change in $T_m$ was then plotted against increasing ligand concentration. The dissociation constants were determined by analyzing the data under non-linear regression (curve fit) model (least-squares ordinary fit) using the binding saturation "one site-specific binding" function.

**Drug sensitivity assay**

Minimum inhibitory concentration ($MIC_{90}$) for the inhibitors was determined using a microplate Alamar blue assay (MABA; Franzblau, 2000). The Alamar blue assay was performed in a sterile 96-well flat-bottom transparent plates. *Mtb* strains were cultured in 7H9 medium supplemented with 10% OADC and grown till exponential phase ($OD_{600} = 0.4$). Approximately $10^5$ bacteria were taken per well in a total volume of 200 µl of media. Auto-fluorescence control was set for the control wells with no *Mtb*. Additional controls were taken for the wells with *Mtb* cells without inhibitor and the wells with only inhibitor and media. After 5-day incubation at 37°C, 20 µl of 10× Alamar blue (Kinetic blue—Krishgen Biosystems—cat. no. CC1100) was added to each well, and the plates were re-incubated for 24 h. The fluorescence intensity was then recorded in a SpectraMax M3 plate reader (Molecular Device) in top-reading mode with excitation at 530 nm and emission at 590 nm. Percentage inhibition was calculated based on the change in relative fluorescence units with increasing inhibitor concentration. The minimum inhibitory concentration that resulted in at least 90% inhibition was identified as $MIC_{90}$.

**Mtb internalization and cell death assay by FACS**

For FACS experiments, a GFP-expressing strain of *Mtb* was used to infect macrophages in similar manner as described above except that $10^5$ cells were seeded per well in a 24-well plate. The adherent macrophages were treated with trypsin, washed twice with PBS, and analyzed with flow cytometry using BD FACSAria FUSION (BD). The samples were excited at 405 and 488 nm lasers at a constant emission (510 nm). The program BD FACS suite software was used to analyze flow cytometry data. For propidium iodide (PI; Sigma cat. no. P4170) staining, macrophages harvested after trypsinization were washed with PBS and incubated with PI (1 µg/ml) dye for 30 s and subjected to flow cytometer.

**Apoptosis detection assay**

*Caspase 3 assay*
The caspase 3/7 levels in the supernatant were determined by using Caspase-Glow® 3/7 Assay reagent (Promega G8091). The macrophages (THP1 and Raw264.7) were seeded in 96-well plate and infected with *Mtb* H37Rv as described earlier at the MOI of 2. The

infected cells were treated with two different concentrations of PRK and incubated for varying time points till 48 h. Post-incubation, the supernatant was collected and incubated with Caspase-Glow 3/7 assay reagent (1:1) for 1 h at 37°C. Post-incubation, the wells were subjected to chemiluminescence and the intensity was recorded using a multi-channel ELISA plate reader (Thermo Scientific—Verioscan reader). All the experiments were done in triplicate and plotted using GraphPad Prism-7 software.

*Arginine supplementation assay*
*Mtb* H37Rv cells were incubated in 7H9 media with only 1% of OADC. The secondary culture was divided into four sets and each one treated with either PRK (1 µg/ml), PRK + arginine (1 mM), Rif (0.4 µg/ml), or Rif + arginine (1 mM) for 24 h. The cells were then washed thrice with PBS and plated on 7H10 agar supplemented with 1% ADC at three different dilutions. The CFU counts were calculated based on the varying dilution of the same sample.

**Mass spectrometric analysis**

*Mtb* H37Rv cells were grown as mentioned, and a secondary culture of 100 ml was incubated in a roller incubator chamber in roller bottles. Equal number of cells were taken and divided into five sets, each treated with either DMSO or PRK at four different concentrations (0.1, 0.5, 1, and 5 µg/ml) for 4 h. An internal standard of heavy isotope-labeled ($^{13}C_6$) L-arginine ($m/z$ 180; from Cambridge isotope laboratories) was used as an internal control for extraction efficiency and comparability in the metabolite levels from different conditions. After 4-h treatment, cells were harvested, washed twice with PBS, and resuspended in the extraction solution ($H_2O$/acetonitrile/methanol in the ratio of 40:40:20) and lysed by bead beating. The extracted metabolite samples were passed through a Sep-PAC column prior to ESI/MS analysis for identification of the arginine levels ($m/z$ 174). Bruker HTC Ultra (ETDII) ESI ion trap instrument was used for mass spectrometry, and samples were passed through C-18 column through a mobile phase of acetonitrile and water with 0.1% formic acid. Flow rate was maintained at 0.2 ml/min, and sample volume injected was 10 µl. The peak was confirmed for arginine by fragmentation pattern analysis (MS/MS of the selected peak).

***Mycobacterium tuberculosis* infection of mice through aerosol and drug treatment**

For chronic model of infection, BALB/c mice were infected via aerosol through Madison chamber aerosol generation instrument calibrated to deliver 100 CFU. Mice were infected with *Mtb* H37Rv strain at 100 CFU of bacilli per mouse. Mice were caged for 4 weeks for establishment of infection. At specific time points (0, 12, and 24 days) post-treatment, mice were sacrificed (cervical dislocation) and their lungs were removed and processed for investigation of bacillary load. Once the infection was established, as determined by bacterial CFU count, mice were divided into five sets, untreated, PBS-treated, Rif-treated (10 mg/kg body weight), PRK-treated (40 mg/kg body weight), and PRK + Rif-treated. The dosage for PRK was determined by the $MIC_{90}$ obtained in the cell-based studies. The pharmacokinetics of PRK is already reported in various studies and is safe for IP administration (Asano *et al*, 2009; Ye *et al*, 2017). The clearance rate from the body is also reported to be very

efficient and safe. Each group contained 18 mice, 6 sacrificed at 0 day of treatment, 6 at 12th day of treatment and 6 at 24th day of treatment. CFUs were determined by homogenizing the lungs and plating appropriate serial dilutions (10-, 100-, and 1,000-fold) on 7H11 supplemented with OADC and PANTA (BD-245114) plates. Colonies were observed and counted after 21 days of incubation at 37°C. Lungs were analyzed for the formation of tubercular granulomas in treated versus untreated conditions. Histopathology analysis was performed as described previously (Singh *et al*, 2003). Briefly, sections of lungs, spleen, and liver were fixed in 10% neutral buffered formalin for embedding in paraffin, sectioning, and staining with hematoxylin and eosin (H&E). A blinded examination of at least three serial sections from each mouse was carried out to evaluate the number of granulomas (lungs) and tissue-associated histopathology (spleen and liver).

All the animal experiments were performed strictly according to the guidelines of the animal ethics committee (AEC). A license to perform the mentioned animal experiments was taken from the institute's AEC well in advance before the commencement of the animal experiments. All the mice used were BALB/c female mice, aged 3–4 weeks at the beginning of the experiment. Animals were caged throughout the experiment in well-ventilated cages in clean rooms of BSL3 facility, in complete isolation. Once humanely sacrificed for the study by cervical dislocation, the carcasses were double-autoclaved and incinerated separately in an incineration chamber.

### qPCR primer sequence

5-LO_Fwd: 5′ CTACGATGTCACCGTGGATG 3′
5-LO_Rev: 5′ GTGCTGCTTGAGGATGTGAA 3′
COX2_Fwd: 5′ GCTCAAACATGATGTTTGCATTC 3′
COX2_Rev: 5′ GCTGGCCCTCGCTTATGA 3′
FLAP_Fwd: 5′ TCTACACTGCCAACCAGAAC 3′
FLAP_Rev: 5′ ACGGACATGAGGAACAGG 3′
CYSLTR1_Fwd: 5′ GGT GCT GAG GTA CCA GAT AG 3′
CYSLTR1_Rev: 5′ CAT GTT CTC CAG GAA TGT CT 3′
MCP1_Fwd: 5′ GGA GCA TCC ACG TGT TGG C 3′
MCP1_Rev: 5′ ACA GCT TCT TTG GGA CAC C 3′

### Statistical analysis

All the data were derived from at least three independent experiments. Statistical analyses were conducted using GraphPad Prism software, and values were presented as mean ± SD. The statistical significance of the differences between experimental groups was determined by two-tailed, unpaired Student's *t*-test unless specified. Differences with a *P*-value of < 0.01 were considered significant. An account of the *P*-values and "*n*" for all the experiments are given in Appendix Table S7.

Rest of the methods involved in this study are given in detail in the Appendix information provided.

**Expanded View** for this article is available online.

### Acknowledgements
We thank Dr. Amit Singh and his laboratory members for the support provided during BSL-3 experiments, essential *Mtb* strains, and plasmid along with valuable discussions. We thank Dr. Ankur Sharma for valuable discussions. We thank Vasista Adiga from the CIDR FACS facility at IISc for flow cytometry experiments. We thank Ms. Sunita from proteomics facility of MBU, IISc, for her help in mass spectrometry. We acknowledge DBT-IISc-supported BSL3 facility for carrying out experiments on *Mtb* strains. The study is supported by grant from Department of Science and Technology (DST) and partially from the Council of Scientific and Industrial Research (CSIR), Indian Council of Medical Research (ICMR), and Department of Biotechnology (DBT) to AS. AS is Bhatnagar fellow, CSIR. AM is supported by CSIR-SRF. ASM is supported by DST-Young Scientist fellowship.

## Author contributions
AM and AS conceived and designed the study. AS provided the entire infrastructure and supported the research. AM performed experiments and analyzed the data. AM and AS wrote the manuscript. AS corrected the manuscript. ASM performed and analyzed computational studies. RSR and AM performed animal experiments. AR and RR helped with the biochemical experiments and microbiology. All authors reviewed the results and approved the final version of the manuscript.

## Conflict of interest
The authors declare that they have no conflict of interest.

# References

Aguiló N, Uranga S, Marinova D, Martín C, Pardo J (2014) Bim is a crucial regulator of apoptosis induced by *Mycobacterium tuberculosis*. *Cell Death Dis* 5: e1343

---

### The paper explained

**Problem**
*Mycobacterium tuberculosis (Mtb)*, the causative agent of tuberculosis (TB), is one of the deadliest infections, in humans worldwide. Immunodeficient individuals are most susceptible to TB and it remains one of the major cause of death in HIV patients. The existing treatment regime against *tuberculosis* is not adequate, and novel therapeutic advancements are required to target *Mtb* pathogenesis.

**Results**
We have identified Pranlukast (PRK) as an inhibitor of arginine biosynthesis in *Mtb*. We show that PRK targets a unique arginine biosynthesis enzyme, exclusive to *Mtb,* thereby impeding its arginine production. PRK efficiently reduces the *Mtb* survival both *in vitro* and within macrophages. Interestingly, PRK also inhibits the 5-lipoxygenase pathway in the macrophages infected with *Mtb*, a pathway that helps the pathogen to survive within the host. This leads to enhanced potency of PRK on the macrophage-internalized *Mtb,* thereby acting as a dual-edged sword. We then demonstrate that PRK works best in combination with the standard-of-care TB therapy drugs. Further, in mice models of *Mtb* infection, PRK alone and in combination with rifampicin efficiently reduce the lung-associated tubercular granulomas and bacterial burden without affecting the host.

**Impact**
This is the first report that shows PRK as a potential drug against *Mtb*. The enhanced efficacy of PRK on the host infection models and its augmented response in combination with the therapy drugs attest to its promise as a potential drug against tuberculosis. Since PRK is an FDA-approved molecule, our study shows the enormous potential of its direct repurposing for novel anti-TB therapeutics.

Asano K, Nakade S, Shiomi T, Nakajima T, Suzuki Y, Fukunaga K, Oguma T, Sayama K, Fujita H, Tanigawara Y *et al* (2009) Impact of pharmacokinetics and pharmacogenetics on the efficacy of pranlukast in Japanese asthmatics. *Respirology* 14: 822–827

Barnes NC, Pujet JC (1997) Pranlukast, a novel leukotriene receptor antagonist: results of the first European, placebo controlled, multicentre clinical study in asthma. *Thorax* 52: 523–527

Behar SM, Divangahi M, Remold HG (2010) Evasion of innate immunity by *Mycobacterium tuberculosis*: is death an exit strategy? *Nat Rev Microbiol* 7: 845

Behar SM, Martin CJ, Booty MG, Nishimura T, Zhao X, Gan H-X, Divangahi M, Remold HG (2011) Apoptosis is an innate defense function of macrophages against *Mycobacterium tuberculosis*. *Mucosal Immunol* 4: 279–287

Boshoff HIM, Myers TG, Copp BR, McNeil MR, Wilson MA, Barry CE (2004) The transcriptional responses of *Mycobacterium tuberculosis* to inhibitors of metabolism: novel insights into drug mechanisms of action. *J Biol Chem* 279: 40174–40184

Capodagli GC, Sedhom WG, Jackson M, Ahrendt KA, Pegan SD (2014) A noncompetitive inhibitor for *Mycobacterium tuberculosis*'s class IIa fructose 1,6-bisphosphate aldolase. *Biochemistry* 53: 202–213

Cardamone M, Puri NK (1992) Spectrofluorimetric assessment of the surface hydrophobicity of proteins. *Biochem J* 282: 589–593

Cole ST, Brosch R, Parkhill J, Garnier T, Churcher C, Harris D, Gordon SV, Eiglmeier K, Gas S, Barry CE *et al* (1998) Deciphering the biology of *Mycobacterium tuberculosis* from the complete genome sequence. *Nature* 393: 537–544

Derrick SC, Morris SL (2007) The ESAT6 protein of *Mycobacterium tuberculosis* induces apoptosis of macrophages by activating caspase expression. *Cell Microbiol* 9: 1547–1555

Divangahi M, Desjardins D, Nunes-Alves C, Remold HG, Behar SM (2010) Eicosanoid pathways regulate adaptive immunity to *Mycobacterium tuberculosis*. *Nat Immunol* 11: 751–758

Drazen JM (1998) Leukotrienes as mediators of airway obstruction. *Am J Respir Crit Care Med* 158: S193–S200

Duan L, Gan H, Golan DE, Remold HG (2002) Critical role of mitochondrial damage in determining outcome of macrophage infection with *Mycobacterium tuberculosis*. *J Immunol* 169: 5181–5187

Escudier B, Eisen T, Stadler WM, Szczylik C, Oudard S, Siebels M, Negrier S, Chevreau C, Solska E, Desai AA *et al* (2007) Sorafenib in advanced clear-cell renal-cell carcinoma. *N Engl J Med* 357: 203

Franzblau S (2000) A rapid, microplate-based assay for evaluating the activity of drugs against *Mycobacterium leprae*, employing the reduction of Alamar Blue. *Lepr Rev* 71: S74–S75

Gordhan BG (2002) Construction and phenotypic characterization of an auxotrophic mutant of *Mycobacterium tuberculosis* defective in L-arginine biosynthesis. *Infect Immun* 70: 3080–3084

Ichiyama T, Hasegawa M, Ueno Y, Makata H, Matsubara T, Furukawa S (2005) Cysteinyl leukotrienes induce monocyte chemoattractant protein 1 in human monocytes/macrophages. *Clin Exp Allergy* 35: 1214–1219

Ichiyama T, Hasegawa M, Hashimoto K, Matsushige T, Hirano R, Furukawa S (2009) Cysteinyl leukotrienes induce macrophage inflammatory protein-1 in human monocytes/macrophages. *Int Arch Allergy Immunol* 148: 147–153

Iyer D, Vartak SV, Mishra A, Goldsmith G, Kumar S, Srivastava M, Hegde M, Gopalakrishnan V, Glenn M, Velusamy M *et al* (2016) Identification of a novel BCL2-specific inhibitor that binds predominantly to the BH1 domain. *FEBS J* 283: 3408–3437

Koul A, Arnoult E, Lounis N, Guillemont J, Andries K (2011) The challenge of new drug discovery for tuberculosis. *Nature* 469: 483–490

Lipinski CA, Lombardo F, Dominy BW, Feeney PJ (1997) Experimental and computational approaches to estimate solubility and permeability in drug discovery and development settings. *Adv Drug Deliv Rev* 23: 3–25

Makarov V, Manina G, Mikusova K, Mollmann U, Ryabova O, Saint-Joanis B, Dhar N, Pasca MR, Buroni S, Lucarelli AP *et al* (2009) Benzothiazinones kill *Mycobacterium tuberculosis* by blocking arabinan synthesis. *Science* 324: 801–804

Makarov V, Lechartier B, Zhang M, Neres J, van der Sar AM, Raadsen SA, Hartkoorn RC, Ryabova OB, Vocat A, Decosterd LA *et al* (2014) Towards a new combination therapy for tuberculosis with next generation benzothiazinones. *EMBO Mol Med* 6: 372–383

Marc F, Weigel P, Legrain C, Glansdorff N, Sakanyan V (2001) An invariant threonine is involved in self-catalyzed cleavage of the precursor protein for ornithine acetyltransferase. *J Biol Chem* 276: 25404–25410

Mayer-Barber KD, Andrade BB, Oland SD, Amaral EP, Barber DL, Gonzales J, Derrick SC, Shi R, Kumar NP, Wei W *et al* (2014) Host-directed therapy of tuberculosis based on interleukin-1 and type I interferon crosstalk. *Nature* 511: 99–103

Mdluli K, Spigelman M (2006) Novel targets for tuberculosis drug discovery. *Curr Opin Pharmacol* 6: 459–467

Neres J, Pojer F, Molteni E, Chiarelli LR, Dhar N, Boy-Rottger S, Buroni S, Fullam E, Degiacomi G, Lucarelli AP *et al* (2012) Structural basis for benzothiazinone-mediated killing of *Mycobacterium tuberculosis*. *Sci Transl Med* 4: 150ra121

Niesen FH, Berglund H, Vedadi M (2007) The use of differential scanning fluorimetry to detect ligand interactions that promote protein stability. *Nat Protoc* 2: 2212–2221

Palde PB, Bhaskar A, Pedró Rosa LE, Madoux F, Chase P, Gupta V, Spicer T, Scampavia L, Singh A, Carroll KS (2016) First-in-class inhibitors of sulfur metabolism with bactericidal activity against non-replicating *M. tuberculosis*. *ACS Chem Biol* 11: 172–184

Ramakrishnan L (2012) Revisiting the role of the granuloma in tuberculosis. *Nat Rev Immunol* 37: 26–366

Renaud J-P, Chung C-W, Danielson UH, Egner U, Hennig M, Hubbard RE, Nar H (2016) Biophysics in drug discovery: impact, challenges and opportunities. *Nat Rev Drug Discov* 15: 679–698

Sankaranarayanan R, Cherney MM, Garen C, Garen G, Niu C, Yuan M, James MNG (2010) The molecular structure of ornithine acetyltransferase from *Mycobacterium tuberculosis* bound to ornithine, a competitive inhibitor. *J Mol Biol* 397: 979–990

Sassetti CM, Boyd DH, Rubin EJ (2003) Genes required for mycobacterial growth defined by high density mutagenesis. *Mol Microbiol* 48: 77–84

Sassetti CM, Rubin EJ (2003) Genetic requirements for mycobacterial survival during infection. *Proc Natl Acad Sci USA* 100: 12989–12994

Singh R, Rao V, Shakila H, Gupta R, Khera A, Dhar N, Singh A, Koul A, Singh Y, Naseema M *et al* (2003) Disruption of mptpB impairs the ability of *Mycobacterium tuberculosis* to survive in guinea pigs. *Mol Microbiol* 50: 751–762

Theron AJ, Steel HC, Tintinger GR, Gravett CM, Anderson R, Feldman C (2014) Cysteinyl leukotriene receptor-1 antagonists as modulators of innate immune cell function. *J Immunol Res* 2014: 608930–16

Tran AT, Watson EE, Pujari V, Conroy T, Dowman LJ, Giltrap AM, Pang A, Wong WR, Linington RG, Mahapatra S *et al* (2017) Sansanmycin natural product analogues as potent and selective anti-mycobacterials that inhibit lipid I biosynthesis. *Nat Commun* 8: 14414

Wenthur CJ, Gentry PR, Mathews TP, Lindsley CW (2014) Drugs for allosteric sites on receptors. *Annu Rev Pharmacol Toxicol* 54: 165–184

WHO (2012) *Global tuberculosis report 2012.* Geneva: World Health Organization

Xu Y, Labedan B, Glansdorff N (2007) Surprising arginine biosynthesis: a reappraisal of the enzymology and evolution of the pathway in microorganisms. *Microbiol Mol Biol Rev* 71: 36–47

Ye X-L, Lu L-Q, Li W, Lou Q, Guo H-G, Shi Q-J (2017) Oral administration of ampelopsin protects against acute brain injury in rats following focal cerebral ischemia. *Exp Ther Med* 13: 1725–1734

Zumla A, Nahid P, Cole ST (2013) Advances in the development of new tuberculosis drugs and treatment regimens. *Nat Rev Drug Discovery* 12: 388–404

Zuniga ES, Early J, Parish T (2015) The future for early-stage tuberculosis drug discovery. *Future Microbiol* 10: 217–229

