## [Review Process File · EMBO Molecular Medicine]

An allosteric inhibitor of *Mycobacterium tuberculosis* ArgJ: Implications to a novel combinatorial therapy

Archita Mishra, Ashalatha S. Mamidi, RS Rajmani, Ananya Ray, Rajanya Roy, Avadhesh Surolia

Review timeline:	Submission date:	16 May 2017
	Editorial Decision:	09 June 2017
	Appeal received:	16 June 2017
	Editorial Decision:	23 June 2017
	Revision received:	22 November 2017
	Editorial Decision:	10 January 2018
	Revision received:	19 January 2018
	Accepted:	26 January 2018

Editor: Céline Carret

Transaction Report:

1st Editorial Decision

09 June 2017

Thank you for the submission of your manuscript "Identification of selective allosteric inhibitor of *M. tuberculosis* ArgJ: Implications towards novel combinatorial therapy". We have now heard back from the two referees whom we asked to evaluate your manuscript.

As you will see, the referees acknowledge the potential pharmacological interest of the study. However, both are also concerned about the in vivo MOA relevance and raised a number of serious conceptual and experimental shortcomings of the study. They feel that given these limitations, your conclusions appear to not be fully supported by the data. As clear and conclusive insight into a novel clinically relevant observation is key for publication in EMBO Molecular Medicine, and together with the fact that we only accept papers that receive enthusiastic support upon initial review, I am afraid that we cannot offer to consider the manuscript further.

I am sorry to have to disappoint you on this occasion, and hope nevertheless that these comments will prove constructive as you prepare the manuscript for submission to another journal.

***** Reviewer's comments *****

Referee #1 (Comments on Novelty/Model System):

detailed notes in critique for the authors.

Referee #1 (Remarks):

Review of EMM-2017-08038 "Identification of selective allosteric inhibitor of *M. tuberculosis* ArgJ: Implications towards novel combinatorial therapy" by Prof. Surolia et al.

This manuscript describes studies that identify putative inhibitors of ArgJ, an essential enzyme involved in arginine biosynthesis. The authors express and purify ArgJ and employ in silico tools to

predict the existence of a potential hydrophobic pocket that they assert is an allosteric regulatory site at the interface of four loops in the protein structure. They then look at binding of a hydrophobic dye, ANS, to the protein and show an increase in fluorescence upon incubation with the protein. ANS suppressed enzymatic activity slightly, which is the primary evidence this is an allosteric regulatory site. They then perform in silico screening of a small collection of approved drugs and test some of these as enzyme inhibitors to arrive at their two leads PRK and SRB. They then show that these two molecules have growth inhibitory activity against MTb in vitro, in macrophages and in mice.

On balance, there are some very interesting observations in this study, and the activity in murine TB is exciting indeed. The major problem with the study, which seems to go without comment, is the large disconnect between the in vitro activity of the two compounds against their supposed target (in the 100s of μM) and the apparent potent activity of these agents in vivo (as low as $0.5\mu\text{g/ml}$ ($1\mu\text{M}$) appears to have some antibacterial activity in infected macrophages. The first obvious conclusion is that the in vivo effect is not likely related in any way to the activity against ArgJ, and when one explores the PRK literature a bit and realizes that this molecule is a potent antagonist of leukotriene receptors (which play a key role in modulating Mtb survival in macrophages (see, for example PMID: 24990750)) one gets very skeptical indeed that the in vivo effect is due to inhibiting arginine biosynthesis. The in vivo results, from MIC to mouse, are quite interesting but they need a solid mechanistic understanding of why they are achieved - what is the mechanism of action in whole cells of MTb and is that the same as the mechanism in infected macrophages? The biochemistry and in silico work is irrelevant if the phenotype is unrelated to ArgJ inhibition. The 200-fold difference between activity against the enzyme and activity in vivo needs to be explained and resolved before this work is ready to be published.

Other comments:

- (1) Abstract line 1 and throughout the intro. The authors emphasize that TB drugs are poorly tolerated because of cross-inhibition of mammalian homologs of the bacterial targets. That is factually incorrect, toxicity with front line agents is rare and mostly because of hepatotoxicity, not related to the bacterial target at all. In fact cross-inhibition is rarely an issue - rifmapicin targets RNA polymerase and shows no significant cross-inhibition of mammalian RNAP, its hard to think of a single example where having a homologous human protein has befuddled a TB drug.
- (2) The two fold reduction in activity with ANS is not very convincing that this represents a genuine allosteric site, that is not much regulation. These data do not constitute direct evidence that ANS binds to the hydrophobic pocket which they identified. More experimental evidence such as a crystal structure of the inhibitor binding to the pocket or mutagenesis of the pocket is required to support that the pocket is an allosteric site of the enzyme and to use it to set up as a docking site for the in-silico study. Also In Fig 2D, the effect of ANS on the catalytic activity is not dose-dependent.
- (3) High consistency of the biochemical data from the TLC-based assay with a densitometric quantification of the reaction product is difficult to believe. It would be useful to show detailed error analysis from genuine biological replicates of this assay from different enzyme preps.
- (4) In general there is a lot of computational work without much objective measure showing that it achieved anything, it would be useful to try to show that the in silico screening enriched for hits compared to an unscreened deck.
- (5) Showing murine efficacy data without PK information for PRK is unacceptable. The key issue as to whether this effect is in fact clinically meaningful is whether approved doses of the drug are even achievable in people. How was that dose even picked? Why was the PRK administered IP???? I thought this drug was orally available?
- (6) The authors mention that MtArgJ is a mono-functional enzyme based on the review paper (Xu et al, 2007) yet the review paper does not contain any evidence for it. If there is any study on the functional characterization of MtArgJ, it needs to be cited.
- (7) The concentration of the active protein was estimated using densitometric analysis of SDS-PAGE profile because they were not able to remove the un-cleaved MtArgJ from the active protein sample.

However, by the gel filtration chromatography, the active tetramer (84kDa) can be separated from the inactive ones. Fig. 1b showed that two cleaved protein stayed as a complex during the affinity purification and this indicates that the active tetramer can be recovered after the gel filtration. This is particularly important when assessing things like hydrophobic dye binding assays wherein the non-native protein may be interfering with these types of assays.

(8) Why did the authors not use a more typical assay for ArgJ - a common and easier way of testing ArgJ activity is to employ ninhydrin to colorimetrically determine the amount of ornithine at 450nm. The catalytic activity of MtArgJ is not fully characterized well enough to interpret their inhibition assay for the in-vitro validation of in-silico predictions. In figure 1c and d, the red dots on the TLC plate need to be defined. Since the inhibition study was also done using TLC-based assay, they need to show that PRK and SRB does not affect TLC reading.

(9) The data points in Figure 3b do not match with the points in Figure 3c. ex) In b, when [N-AcOrn] is 1mM at [PRK]=100 uM, the velocity is approximately 16 ($1/V = .063$). However, in c, at the same condition, $1/V$ is around .25.

(10) The author stated that ligand-binding to the hydrophobic pocket may lead to conformational changes at substrate binding site since the pocket is located at the center of the two substrate-binding pocket. However, close location does not always imply influence on the active site and experimental validation is required for this assertion, perhaps something like CD would provide some evidence of conformational change?

(11) Their kill kinetics assay indicates that PRK and SRB showed marked reduction in Mtb survival in combination with standard-of-care anti-TB drugs in Figure 5f and g. But, they did not test PRK and SRB only to compare the PRK and SRB to the combination.

(12) Methods: biochemical assay protocol needs more details, such as reaction volume, pH, and so on. What is the K_m value for Glutamate?

Referee #2 (Comments on Novelty/Model System):

The authors have not adequately assessed the mechanism by which PRK affects growth of Mtb and appears to kill Mtb inside macrophage-like cells. The synergy with rifampicin in the mouse model is promising.

Referee #2 (Remarks):

Mishra and colleagues identified allosteric inhibitors of Mtb's ornithine acetyltransferase, characterized their whole cell activity in vitro and in macrophage-like cells and showed that they increase the potency of rifampicin in a mouse model of tuberculosis.

These are interesting findings.

My main concern is the lack of demonstration of on-target activity of pranlukast (PRK) when used against live Mtb. This is an important but missing piece of data. While the authors demonstrate that PRK can inhibit ornithine acetyltransferase in vitro, there is no evidence that the whole cell activity occurs via the same target. There are many ways to accomplish this. First and most simply, the authors need to show that supplementation of the growth medium with arginine abolishes the activity of PRK, as arginine biosynthesis become dispensable for Mtb to grow when arginine is provided. The authors can also over- or under-express ArgJ and test if this is associated with a shift of the MIC of PRK.

The authors should determine the minimal bactericidal concentration (MBC) in addition to the MIC of PRK. They show growth inhibition by PRK in vitro and killing of Mtb in THP1 and Raw264.7 cells. It is important to determine if PRK can also kill in vitro and with what MBC or if PRK inside cells synergized with a host defense pathway. Importantly, as mentioned above, the in vitro activity should depend on the lack of arginine in the medium.

It is concerning that PRK has a K_i of 132 uM against purified ArgJ while its MIC against live bacteria was only 5 ug/ml (significantly less than 100 uM). This strongly suggests a different or additional target for PRK in whole cells.

In Figure 5 b & c, the graphs y-axes are mislabeled. The authors are not measuring % survival, the alamar blue assay does not measure bacterial viability. This needs to be changed. The in vivo data are promising, but not well displayed. The authors should show CFU data for the time points before drug treatment and the CFU data for PRK, RIF, RIF+PRK and the vehicle treated control group at all time points. The gross pathology images are not convincing. The authors should show H&E stained microscopic images of lung lobes. I think the conclusion that "PRK treatment reduces lung-associated granuloma from Mtb infected mice" is not justified based on the current images. There is no mentioning of how the authors prevented drug carry over in the mouse and macrophage cell line experiments. This is an important point, and the authors need to ensure that killing occurred inside cells/in vivo and not by residual drug on the agar plates.

Appeal

16 June 2017

We really appreciate the efforts taken by the reviewers and value their constructive comments on the above manuscript. However, I strongly believe that we are in a position to answer all the points raised by the reviewers.

The major concern arising from both the reviewers is about the validation of mechanism of action through which PRK inhibits the Mtb cell survival. We are addressing this part in our lab and the experiments are already in progress. We will be able to demonstrate the direct inhibition of arginine biosynthesis pathway upon inhibitor treatment in Mtb cells.

Also, as pointed out by the first reviewer, we are well aware of the affinity of PRK towards the leukotriene receptor (CysLTR1) and have referenced this in the manuscript as well. There are reports on the role of leukotriene signalling in Mtb survival within the macrophages. Hence, we are already in process of investigating the effect of PRK on the leukotriene signalling in the macrophages, under infected conditions. This will also explain the improved in vivo efficacy of PRK over in vitro results.

Also, as suggested by the reviewers, we are already probing further the mechanism of PRK's interaction with the allosteric pocket on MtArgJ by site-directed mutagenesis to validate the MD simulation results.

Additionally, we now have the histopathological data and H&E staining results of mice organs, along with the precise granuloma score of the treated mice lungs vs untreated mice. These data strengthen our conclusion that "PRK treatment reduces lung associated granuloma in infected mice, without effecting the host".

Also, we have data for the mono-functional nature of MtArgH, as asked by the first reviewer. Since we can satisfactorily address all the points raised by the reviewers, we request you to allow us to submit a suitably revised version of the manuscript together with a point-wise rebuttal of their concerns, in the next 4 months' time.

Given the novelty of the work and the acknowledged interest of the reviewers in the findings of this study, I hope you will find the revised version of the manuscript worthy enough, based on its merit, for resubmission to EMBO Molecular Medicine.

2nd Editorial Decision

23 June 2017

Thank you for your letter claiming that you can and have started addressing all issues raised by the referees on your paper.

I have discussed within the team and we decided to give your paper a chance, thus allow revision of the current work.

I want to stress however, that given the many substantial issues initially raised, there is a strong risk that the referees will not find the revision sufficient for publication. I will make my best to secure the same referees. But as the paper was rejected, they may not agree to re-review.

We would therefore welcome the submission of a revised version within four months for further consideration and would like to encourage you to address **all** the criticisms raised as suggested to

improve conclusiveness and clarity. Please note that EMBO Molecular Medicine strongly supports a single round of revision and that, as acceptance or rejection of the manuscript will depend on another round of review, your responses should be as complete as possible.

1st Revision - authors' response

22 November 2017

(begins on next page)

Response to reviewers

We thank both the reviewers for their constructive comments on the manuscript. In the revised version of manuscript, we have aimed to address all the concerns raised by reviewers. This endeavor has significantly improved the manuscript specially in terms of dissecting the mechanism of *in vivo* action in detail. Below we are providing our detailed response (in blue) to Referee remarks (in red).

Referee #1 (Remarks):

Review of EMM-2017-08038 "Identification of selective allosteric inhibitor of M. tuberculosis ArgJ: Implications towards novel combinatorial therapy" by Prof. Suroliya et al. This manuscript describes studies that identify putative inhibitors of ArgJ, an essential enzyme involved in arginine biosynthesis. The authors express and purify ArgJ and employ *in silico* tools to predict the existence of a potential hydrophobic pocket that they assert is an allosteric regulatory site at the interface of four loops in the protein structure. They then look at binding of a hydrophobic dye, ANS, to the protein and show an increase in fluorescence upon incubation with the protein. ANS suppressed enzymatic activity slightly, which is the primary evidence this is an allosteric regulatory site. They then perform *in silico* screening of a small collection of approved drugs and test some of these as enzyme inhibitors to arrive at their two leads PRK and SRB. They then show that these two molecules have growth inhibitory activity against MTb *in vitro*, in macrophages and in mice.

On balance, there are some very interesting observations in this study, and the activity in murine TB is exciting indeed. The major problem with the study, which seems to go without comment, is the large disconnect between the *in vitro* activity of the two compounds against their supposed target (in the 100s of μM) and the apparent potent activity of these agents *in vivo* (as low as 0.5 $\mu\text{g}/\text{ml}$ (1 μM) appears to have some antibacterial activity in infected macrophages.

The first obvious conclusion is that the *in vivo* effect is not likely related in any way to the activity against ArgJ, and when one explores the PRK literature a bit and realizes that this molecule is a potent antagonist of leukotriene receptors (which play a key role in modulating Mtb survival in macrophages (see, for example PMID: 24990750) one gets very skeptical indeed that the *in vivo* effect is due to inhibiting arginine biosynthesis.

The *in vivo* results, from MIC to mouse, are quite interesting but they need a solid mechanistic understanding of why they are achieved - what is the mechanism of action in whole cells of MTb and is that the same as the mechanism in infected macrophages? The biochemistry and *in silico* work is irrelevant if the phenotype is unrelated to ArgJ inhibition. The 200-fold difference between activity against the enzyme and activity *in vivo* needs to be explained and resolved before this work is ready to be published.

Author's (Response):

We thank the reviewer for the appreciation of our study together with the constructive comments which have helped us in improving the manuscript. Reviewer has made an interesting observation regarding the elevated *in vivo* efficiency. First and foremost, we aimed to address the mechanism of action of PRK in *Mtb* cells. In

revised manuscript, we have explored the mechanistic explanation for enhanced *in vivo* efficiency along with data suggesting PRK impedes *in vivo* Arginine biosynthesis in *Mtb* cells. A point wise response to the questions raised by the reviewer is given below.

One of the major concern was the lack of direct evidence that the whole cell activity of Pranlukast (PRK) occurs via the same target. We have addressed this concern by two different methods:

A) To address whether PRK indeed inhibits arginine biosynthesis *in vivo* we performed the rescues experiment by arginine supplementation assay. We show that supplementing the *Mtb* culture with arginine could rescue the PRK mediated cell death. The result clearly demonstrates the inhibitors specificity for arginine biosynthesis (Rebuttal letter Fig. 1a and revised manuscript Fig. 7a). Moreover, addition of arginine had no effect on Rifampicin mediated killing of *Mtb* cells. This shows that PRK induces its bactericidal effect on *Mtb* by reducing the arginine levels in the pathogen.

B) We also investigated the effect of PRK treatment on arginine levels by Mass spectrometry. We treated the *Mtb* cultures with PRK and the whole cell metabolites were isolated and subjected to ESI-MS analysis. The area and intensity of the arginine peak (m/z 174) was determined and the arginine levels were estimated based on a standard arginine plot. A heavy isotope labelled arginine (^{13}C labelled arginine: m/z = 180) was used as an internal standard. We observed a steady decline in the arginine levels as shown below, at increasing PRK concentration (Rebuttal letter Fig. 1b and revised manuscript Fig. 7b). These results support arginine supplementation assay and show that the effect of PRK is indeed through the inhibition of arginine biosynthesis.

Rebuttal Fig. 1 PRK targets the arginine biosynthesis in *Mtb* (a) CFU analysis of *Mtb* H37Rv treated with DMSO control, PRK, PRK with arginine supplement, Rif and Rif with arginine supplement. As observed, arginine supplementation could reverse the effect of PRK on *Mtb*. As a control, arginine supplementation had no effect on the Rifampicin mediated cell death. (b) ESI/MS analysis of the whole cell metabolites isolated from PRK treated *Mtb* samples. DMSO treated samples were taken as control. Concentration of arginine in each sample was calculated by a standard arginine plot (inset).

The reviewer has also raised an interesting observation that the *in vivo* efficacy of the drug in the macrophage model of infection could be due to simultaneous effect of PRK on the host and MTb. We have demonstrated the explanation for the same in the revised version.

PRK is a known inhibitor of CysLTR1 which is involved in leukotriene signalling, essential for *Mtb* pathogenesis. However, studies have shown that in macrophages and dendritic cells, PRK also acts on the lipoxygenase pathway, thereby targeting the eicosanoid (**leukotriene** and prostaglandin) biosynthesis, which are ligands for CysLTR's (Theron *et al*, 2014). Eicosanoids are synthesized by enzymes like 5-LO (5-lipoxygenase), FLAP (5-lipoxygenase-activating protein) and COX-2 (cyclooxygenase-2) and have been associated with tuberculosis pathogenesis (Mayer-Barber *et al*, 2014; Drazen, 1998). Interestingly, Divangahi *et al*. have shown that *M. tuberculosis* infection activates the 5-lipoxygenase (5-LO) pathway in the host macrophages, which facilitates the pathogen survival within the host (Divangahi *et al*, 2010; Behar *et al*, 2010).

We hypothesized that PRK's enhanced efficacy *in vivo*, could be due to combinatorial targeting of leukotriene signalling (in macrophage) and Arginine biosynthesis (in *Mtb*). Firstly, we demonstrate that CysLTR1 and 5-lipoxygenase (5-LO) genes are significantly upregulated (by 4.5-fold and 3.5-fold respectively), in the *Mtb* infected macrophages (*Rebuttal letter Fig. 2c and revised manuscript Fig. 7c*). Along with that FLAP and COX-2, other mediators of the eicosanoid biosynthesis, were also upregulated by *Mtb* infection (*Rebuttal letter Fig. 2c and revised manuscript Fig. 7c*).

Interestingly, upon PRK treatment, we observed a remarkable decline in the transcript levels of 5-LO (4.5-fold), Cox-2 (8.5-fold) and FLAP (2.5-fold) (*Rebuttal letter Fig. 2d-f and revised manuscript Fig. 7d-f*). This shows that PRK downregulates the 5-lipoxygenase pathway, thereby reducing the *Mtb* survival and dissemination within the macrophages.

We also show that PRK treatment causes a prominent decline in the CysLTR1 transcript levels in the *Mtb* infected macrophages, along with downregulation in the MCP-1 levels, a down-stream effector of CysLTR1 signalling (Ichiyama *et al*, 2009; 2005) (*Rebuttal letter Fig. 2g, h and revised manuscript Fig. 7g-h*).

Taken together, these results demonstrate the combinatorial effect of PRK on *Mtb* infected macrophages. In addition to targeting the arginine biosynthesis in the pathogen, PRK also inhibits the host 5-lipoxygenase signalling, thereby potentiating its bactericidal effect in the macrophage model of infection. This study also sheds light on the importance of lipoxygenase signaling and leukotrienes in *Mtb* pathogenesis.

We sincerely thank reviewer for probing us on this issue which lead to new mechanistic insights and better understanding of PRK's on inhibiting MTb infection in macrophage model.

Rebuttal Fig. 2: PRK targets 5-lipoxygenase signaling in the *Mtb* infected host macrophages. (c) Q-PCR analysis of the genes CysLTR1, 5-LO, FLAP, COX-2 and MCP1 in the *Mtb* infected vs uninfected macrophages (Raw 264.7). Q-PCR analysis of the genes (d) 5-LO, (e) COX-2, (f) FLAP, (g) CysLTR1 and (h) MCP1 upon PRK treatment in the infected macrophages, DMSO and Rif treatment as controls. As observed, 5-lipoxygenase and the associated genes involved in eicosanoid biosynthesis were significantly down-regulated upon PRK treatment but not by Rif treatment.

Other comments:

(1) Abstract line 1 and throughout the intro. The authors emphasize that TB drugs are poorly tolerated because of cross-inhibition of mammalian homologs of the bacterial targets. That is factually incorrect, toxicity with front line agents is rare and mostly because of hepatotoxicity, not related to the bacterial target at all. In fact, cross-inhibition is rarely an issue - rifampicin targets RNA polymerase and shows no significant cross-inhibition of mammalian RNAP, its hard to think of a single example where having a homologous human protein has befuddled a TB drug.

We thank the reviewer for pointing out the mistake and we have corrected this throughout the text.

(2) The two-fold reduction in activity with ANS is not very convincing that this represents a genuine allosteric site, that is not much regulation. These data do not constitute direct evidence that ANS binds to the hydrophobic pocket which they identified. More experimental evidence such as a crystal structure of the inhibitor binding to the pocket or mutagenesis of the pocket is required to support that the

pocket is an allosteric site of the enzyme and to use it to set up as a docking site for the in-silico study. Also, In Fig 2D, the effect of ANS on the catalytic activity is not dose-dependent.

We agree with the reviewer's concern, that ANS binding *in vitro* may not prove that it is binding to this newly identified pocket. However, we have also done blind docking (*in silico*) experiments of ANS with the whole molecule of *MtArgJ* and 2000 final conformations were generated. However, we observed that all of the ANS conformers docked inside this novel pocket and none to the substrate binding pocket or any other pockets on the *MtArgJ* surface. This indicates that ANS binds to this large pocket on *MtArgJ*. The result was further confirmed biochemically by the ANS binding assay (*manuscript Fig. 2b, c*). Also, our dye displacement assay (*manuscript Fig. 2 g, h*) demonstrates that a ligand specifically selected for its binding to the allosteric pocket on *MtArgJ* (PRK and SRB, in this case), could displace the bound ANS from the protein surface. These results further confirm for the binding of ANS to the pocket identified by us in this study.

We agree that the two-fold inhibition observed in the *MtArgJ* activity by ANS binding is not sufficient to consider the pocket to be allosteric. However, this data just gave us an indication for the ability of this pocket to alter the active site. It is expected that ANS being a very small and nonspecific molecule will have only marginal effect on the enzyme activity. But this gave us leads to further probe this pocket with a specific molecule to have better enzyme inhibition. Our enzyme inhibition and competition experiments (*manuscript Fig. 3 a, b and e, f*), also confirmed that the inhibitors binding (both PRK and SRB) to the allosteric pocket alters the enzyme activity along with displacing ANS from that pocket.

We agree that PRK-ArgJ co-crystallization will provide the precise information about the inhibitor binding pocket. We also attempted to obtain the crystal structure of the inhibitor bound complex but so far we are unable to crystallize the complex. However, we have addressed this question in the revised version by using site-directed mutagenesis approach. Based on the *in silico* analysis, we selectively mutated specific residues (Gln305, Ser310 and Asp234) from the inhibitor binding pocket. In concordance with the *in silico* predications these mutations lead to the loss of enzymatic activity of *MtArgJ* (Revised manuscript Appendix Fig. S6.B). These results clearly indicate the importance of this pocket as an allosteric site.

(3) High consistency of the biochemical data from the TLC-based assay with a densimetric quantification of the reaction product is difficult to believe. It would be useful to show detailed error analysis from genuine biological replicates of this assay from different enzyme preps.

All the experiments are done in triplicate and are confirmed with at least three biological replicates (different enzyme preparation). We initially plotted the average of the triplicates. As suggested by the reviewer, we have incorporated the error bars in the enzyme kinetics experiments for separate enzyme preparations in the revised version. This did not lead to any significant change in the kinetic parameters.

TLC is a fairly sensitive assay system to study the enzymatic activity and we have repeated the assay for each concentration multiple times with different enzyme preparation, hence standardizing the sensitivity of our assay.

(4) In general, there is a lot of computational work without much objective measure showing that it achieved anything, it would be useful to try to show that the *in silico* screening enriched for hits compared to an unscreened deck.

We understand the reviewer's concern and based on that we have removed the computation based network analysis part from the Appendix in the revised manuscript. However, rest of the computational data is very integral to this study. *In silico* screening of FDA approved Drugs (1556 drug molecules) filtered out most of the non-binders (1513 molecules were sampled out) and gave us a set of 43 ligands for biochemical screen. Upon identifying the hit compounds by *in vitro* screen, classical MD simulations helped in analyzing the molecular interactions involved. This was further confirmed by mutagenesis thereby showing that the computational studies have complemented the experimental findings throughout.

Importantly, we have performed a negative validation of *in silico* screening strategy. None of the compounds tested from the negative deck could inhibit the *MtArgJ* activity, *in vitro* further validating our screening approach. This data is already present in the Appendix section of the manuscript (Appendix Table 4).

(5) Showing murine efficacy data without PK information for PRK is unacceptable. The key issue as to whether this effect is in fact clinically meaningful is whether approved doses of the drug are even achievable in people. How was that dose even picked? Why was the PRK administered IP???? I thought this drug was orally available?

Since PRK is already an FDA approved drug, and is being used in many parts of the world, esp. Japan, its pharmaco-kinetics has been reported by various studies and is proved to be suitable for both children as well as adults (ASANO *et al*, 2009). The studies have administered of upto 1gm/day of Pranlukast which is totally safe in human and has about 95% clearance within 24 hr. The dosage in our study was determined by precise calculations based on the MIC obtained from the *in vitro* experiments (on *Mtb* cells).

In the revised version, we have incorporated the histopathology data (H&E staining) of mice lungs, spleen and hepatic tissues (Appendix Fig S10 & S11). Notably, there was no significant effect on the gross pathology of the hepatic tissues or spleen at the administered dosages. This confirms that there was no toxicity associated with the administered dose in mice.

The reason for IP administration are:

- a). Most of the studies in the past, have used IP route only to administer Pranlukast in the mice models. Also, the patent involving the effect of Pranlukast on inflammation and sepsis too has used IP route for administration of this drug in the (U.S. Patent Application No. 15/269,824).
- b). It is known that the IP route gets substances into the circulation faster than oral route and requires lesser amount of drug sample (Ye *et al*, 2017).

(6) The authors mention that *MtArgJ* is a mono-functional enzyme based on the review paper (Xu *et al*, 2007) yet the review paper does not contain any evidence for

it. If there is any study on the functional characterization of MtArgJ, it needs to be cited.

Yes, the reviewer is correct, that review article didn't have any biochemical evidence to support that. Hence, we also tested this in our laboratory as we started working on MtArgJ and found it to be monofunctional. It was not able to catalyze the transfer of acetyl moiety from the acetyl-coA to glutamate *in vitro*. This data is not shown in the manuscript but if required we can provide it.

(7) The concentration of the active protein was estimated using densitometric analysis of SDS-PAGE profile because they were not able to remove the un-cleaved MtArgJ from the active protein sample. However, by the gel filtration chromatography, the active tetramer (84kDa) can be separated from the inactive ones. Fig.1b showed that two cleaved protein stayed as a complex during the affinity purification and this indicates that the active tetramer can be recovered after the gel filtration. This is particularly important when assessing things like hydrophobic dye binding assays wherein the non-native protein may be interfering with these types of assays.

The MtArgJ gene is initially translated as a 41kDa protein which gets cleaved and forms two fragments of 20kDa and 21kDa each, which then form a heterodimer. Two such heterodimers form an active tetramer of about 82kDa. Now the problem with gel filtration is that the un-cleaved protein (Mw 41kDa) also elutes at 84 kDa. We tried to separate both the fragment (20 and 21kDa) and then reconstitute to form the tetramer but the protein was not active. We tried to clone both the fragments separately and then reconstitute them but the protein was still not showing activity. Marc, F *et al.* also showed that the two fragments of ArgJ from *Bacillus stearothermophilus* if separated and reconstituted will not be active (Marc *et al.*, 2001).

(8) Why did the authors not use a more typical assay for ArgJ - a common and easier way of testing ArgJ activity is to employ ninhydrin to colorimetrically determine the amount of ornithine at 450nm. The catalytic activity of MtArgJ is not fully characterized well enough to interpret their inhibition assay for the *in-vitro* validation of *in-silico* predictions. In figure 1c and d, the red dots on the TLC plate need to be defined. Since the inhibition study was also done using TLC-based assay, they need to show that PRK and SRB does not affect TLC reading.

As mentioned by the reviewer, indeed we begin our vogue with typical colorimetric assay for ArgJ with ninhydrin. However, the presence of Glutamate in our reaction sample interfered with the readout as ninhydrin also interacts with glutamate and gives color. Therefore, we devised this TLC based assay to identify the ornithine spot specifically which was then quantified by Multi-gauge densitometry software. In *Figure 1c and d* the red dots on TLC plate is of glutamate, another reaction product of MtArgJ. We have marked this in the revised version Fig. 1c and Appendix Fig. S1B. We have also included the TLC control images of PRK and SRB, showing that they do not have any interaction with the ninhydrin on TLC (Appendix Fig. S1C).

(9) The data points in Figure 3b do not match with the points in Figure 3c. ex) In b, when [N-AcOrn] is 1mM at [PRK]=100 uM, the velocity is approximately 16 ($1/V = .063$). However, in c, at the same condition, $1/V$ is around .25.

We thank the reviewer for pointing out the mistake and the same has been corrected in the revised version (Fig. 3 c-f)

(10) The author stated that ligand-binding to the hydrophobic pocket may lead to conformational changes at substrate binding site since the pocket is located at the center of the two substrate-binding pocket. However, close location does not always imply influence on the active site and experimental validation is required for this assertion, perhaps something like CD would provide some evidence of conformational change?

The statement that “ligand-binding to the hydrophobic pocket may lead to conformational changes at substrate binding site” was actually a speculation based on our initial results and not solely on the fact that the pocket is situated in-between the two active site. However, we understand that this statement will not be correct unless we have strong evidence for conformational changes. Therefore, we have rephrased it as:

“Based on our initial results, we hypothesize that a suitable ligand, bound to this pocket may cause inhibition of the enzymatic activity of the protein.”

(11) Their kill kinetics assay indicates that PRK and SRB showed marked reduction in *Mtb* survival in combination with standard-of-care anti-TB drugs in Figure 5f and g. But, they did not test PRK and SRB only to compare the PRK and SRB to the combination.

We have tested for the PRK's effect on the killing of *Mtb* cells by CFU analysis and the same is now incorporated in the Fig. 3f and g of the revised manuscript. For easy reference, same is also shown in Rebuttal Fig. 3, kindly note the yellow line on the graphs.

Rebuttal Fig. 3: CFU analysis of *Mtb* treated with PRK/SRB alone or a cocktail of Rif, Inh and Emb abbreviated as RHE and compared with a new combination of RH+PRK and RH+SRB, at two concentrations of PRK and SRB (0.1 μ M and 1 μ M) respectively. (**** $p < 0.0001$, *** $p < 0.001$, ** $p < 0.01$, * $p < 0.1$)

(12) Methods: biochemical assay protocol needs more details, such as reaction volume, pH, and so on. What is the K_m value for Glutamate?

We thank the reviewer for pointing out this and in the revised version we have provided an elaborated version of assay protocols with specifics of the experiments throughout the materials and methods section.

Since we are not using the substrate analogues for inhibition, determining the K_m values for both the substrates did not appear to be essential for this study. The reason to determine the kinetics of N-acetyl ornithine (the primary substrate for MtArgJ) was to characterize the enzyme for its functionality and to determine the mode of inhibition by the inhibitors.

Referee #2 (Remarks):

Mishra and colleagues identified allosteric inhibitors of Mtb's ornithine acetyltransferase, characterized their whole cell activity in vitro and in macrophage-like cells and showed that they increase the potency of rifampicin in a mouse model of tuberculosis.

These are interesting findings.

Ques:

My main concern is the lack of demonstration of on-target activity of pranlukast (PRK) when used against live Mtb. This is an important but missing piece of data. While the authors demonstrate that PRK can inhibit ornithine acetyltransferase in vitro, there is no evidence that the whole cell activity occurs via the same target. There are many ways to accomplish this. First and most simply, the authors need to show that supplementation of the growth medium with arginine abolishes the activity of PRK, as arginine biosynthesis become dispensable for Mtb to grow when arginine is provided. The authors can also over- or under-express ArgJ and test if this is associated with a shift of the MIC of PRK.

Ans:

We thank the reviewer for the appreciation of our study and the constructive comments which have helped us in significantly improving the manuscript. Reviewer has raised some very interesting issues and we have tried to explore them. A point wise response to the questions raised by the reviewer is given below:

One of the major concern was the lack of direct evidence that the whole cell activity of Pranlukast (PRK) occurs via the same target. We have addressed this concern by two different methods:

1) As suggested by the reviewer, we did the arginine supplementation assay. We show that supplementing the *Mtb* culture with arginine could rescue the PRK mediated cell. The result clearly demonstrates the inhibitors specificity for arginine biosynthesis (*Revised manuscript Fig. 7a; rebuttal letter fig 1 a*). Moreover, addition of arginine had no effect on Rifampicin mediated killing of *Mtb* cells. This shows that PRK induces its bactericidal effect on *Mtb* by reducing the arginine levels in the pathogen. Also, the inability of Mtb to survive under arginine deficit created by PRK is also consistent with the suitability of ArgJ as a target against Mtb survival.

2) We also investigate the effect of PRK treatment on arginine levels by Mass spectrometry. We treated the *Mtb* cultures with PRK and the whole cell metabolites were isolated and subjected to ESI-MS analysis. The area and intensity of the arginine peak (m/z 174) was determined and the arginine levels were estimated based on a standard arginine plot. A heavy isotope labelled arginine (¹³C labelled arginine: m/z = 180) was used as an internal standard. We observed a steady decline in the arginine levels as shown below, at increasing PRK concentration.

Rebuttal Fig. 1 PRK targets the arginine biosynthesis in *Mtb* (a) CFU analysis of *Mtb H37Rv* treated with DMSO control, PRK, PRK with arginine supplement, Rif and Rif with arginine supplement. As observed, arginine supplementation could reverse the effect of PRK on *Mtb*. As a control, arginine supplementation had no effect on the Rifampicin mediated cell death. (b) ESI/MS analysis of the whole cell metabolites isolated from PRK treated *Mtb* samples. DMSO treated samples were taken as control. Concentration of arginine in each sample was calculated by a standard arginine plot (inset).

1) The authors should determine the minimal bactericidal concentration (MBC) in addition to the MIC of PRK. They show growth inhibition by PRK in vitro and killing of *Mtb* in THP1 and Raw264.7 cells. It is important to determine if PRK can also kill in vitro and with what MBC or if PRK inside cells synergized with a host defense pathway. Importantly, as mentioned above, the in vitro activity should depend on the lack of arginine in the medium.

We have determined both the MIC₉₀ as well as IC₅₀ for Pranlukast (PRK) and Sorafenib (SRB) by Alamar assay. Since we have calculated the MIC₉₀, which is a measure of minimum drug concentration to attain 90% killing, MBC (minimum drug concentration to attain 99% killing) might not give any new information.

As asked by the reviewer, in the revised version we have incorporated the results of bacterial killing by PRK itself, by the CFU analysis (*Revised manuscript Fig. 5f; rebuttal fig. 2*), also shown below by the yellow line on the graph:

Rebuttal Fig. 2: CFU analysis of *Mtb* treated with PRK/SRB alone or a cocktail of Rif, Inh and Emb abbreviated as RHE and compared with a new combination of RH+PRK and RH+SRB, at two concentrations of PRK and SRB (0.1 μM and 1 μM) respectively. (**** $p < 0.0001$, *** $p < 0.001$, ** $p < 0.01$, * $p < 0.1$)

The reviewer has raised a very important and interesting concern here that PRK may have some synergistic effect *in vivo*, based on its increased potency in the macrophage models of infection. We therefore explored the reason for the enhanced effect of PRK on macrophage internalized *Mtb*. Although PRK is a known inhibitor of CysLTR1, studies have shown that in macrophages and dendritic cells, PRK also acts on the 5-lipoxygenase pathway, thereby targeting the eicosanoid (**leukotriene** and prostaglandin) biosynthesis, which are ligands for CysLTR's (Theron *et al*, 2014). Eicosanoids are synthesized by enzymes like 5-LO (5-lipoxygenase), FLAP (5-lipoxygenase-activating protein) and COX-2 (cyclooxygenase-2) and have been associated with tuberculosis pathogenesis (Drazen, 1998; Mayer-Barber *et al*, 2014). Interestingly, Divangahi *et al*. have shown that *M. tuberculosis* infection activates the 5-lipoxygenase (5-LO) pathway in the host macrophages, which facilitates the pathogen survival within the host (Divangahi *et al*, 2010; Behar *et al*, 2010).

We hypothesized that PRK's enhanced efficacy *in vivo*, could be due to synergistic targeting of leukotriene signalling (in macrophage) and Arginine biosynthesis (in *Mtb*). Firstly, we demonstrate that CysLTR1 and 5-lipoxygenase (5-LO) genes are significantly upregulated (by 4.5-fold and 3.5-fold respectively) in the *Mtb* infected macrophages (*Rebuttal letter Fig. 3 and revised manuscript Fig. 7c*). Along with that FLAP and COX-2, other mediators of the eicosanoid biosynthesis, were also upregulated by *Mtb* infection (*Rebuttal letter Fig. 3 and revised manuscript Fig. 7c*). Interestingly, upon PRK treatment, we observed a remarkable decline in the transcript levels of 5-LO (4.5-fold), Cox-2 (8.5-fold) and FLAP (2.5-fold) (*Rebuttal letter Fig. 3 and Revised manuscript Fig. 7d-f*). This shows that PRK downregulates the 5-lipoxygenase pathway, thereby reducing the *Mtb* survival and dissemination within the macrophages.

We also show that PRK treatment causes a prominent decline in the CysLTR1 transcript levels in the *Mtb* infected macrophages, along with downregulation in the MCP-1 levels, a down-stream effector of CysLTR1 signalling (Ichiyama *et al*, 2009; 2005) (*Rebuttal letter Fig. 3 and Revised manuscript Fig. 7g-h*).

Taken together, these results demonstrate the synergistic effect of PRK on *Mtb* infected macrophages. In addition to targeting the arginine biosynthesis in the pathogen, PRK also targets the host 5-lipoxygenase signalling, thereby potentiating its bactericidal effect in the macrophage model of infection. This study also sheds light on the importance of lipoxygenase signaling and leukotrienes in *Mtb* pathogenesis.

Rebuttal Fig. 3: PRK targets 5-lipoxygenase signaling in the *Mtb* infected host macrophages. (c) Q-PCR analysis of the genes CysLTR1, 5-LO, FLAP, COX-2 and MCP1 in the *Mtb* infected vs uninfected macrophages (Raw 264.7). Q-PCR analysis of the genes (d) 5-LO, (e) COX-2, (f) FLAP, (g) CysLTR1 and (h) MCP1 upon PRK treatment in the infected macrophages, DMSO and Rif treatment as controls. As observed, 5-lipoxygenase and the associated genes involved in eicosanoid biosynthesis were significantly down-regulated upon PRK treatment but not by Rif treatment.

2) It is concerning that PRK has a K_i of 132 μM against purified ArgJ while its MIC against live bacteria was only 5 $\mu\text{g/ml}$ (significantly less than 100 μM). This strongly suggests a different or additional target for PRK in whole cells. In Figure 5 b & c, the graphs y-axes are mislabeled. The authors are not measuring % survival, the alamar blue assay does not measure bacterial viability. This needs to be changed.

We understand that speculation of an additional target of PRK within *Mtb* cannot be completely overruled. However, we would like emphasize that in the revised manuscript, we have shown a decline in the arginine levels upon PRK treatment and supplementing the culture media with arginine can rescue the bactericidal effects of this compound. This clearly shows that the effect of PRK mediated bacterial killing is through the inhibition of arginine biosynthesis in *Mtb*.

The alamar blue assay works as a cell viability indicator by the conversion of resazurin to resorufin, through live cells. The assay has been used in the past in several studies to determine the percentage of viable cells in a culture. In the revised version we have changed the y-axis of the graph (Figure 5 b & c) from percent survival to **cell viability (%)**, a term more commonly used in such assays (Conway *et al*, 2016; Beug *et al*, 2014; Simpson *et al*, 2008; O'Brien *et al*, 2003; Roit *et al*, 2014).

3) The *in vivo* data are promising, but not well displayed. The authors should show CFU data for the time points before drug treatment and the CFU data for PRK, RIF, RIF+PRK and the vehicle treated control group at all time points. The gross pathology images are not convincing. The authors should show H&E stained microscopic images of lung lobes. I think the conclusion that "PRK treatment reduces lung-associated granuloma from *Mtb* infected mice" is not justified based on the current images.

As indicated by the reviewer, we have included the CFU data before and after the drug treatment, for all time points (0 day, 12th day and 24th day) in the revised version of manuscript (Fig. 8 f, g). We have also incorporated the H&E staining images of mice lungs, spleen and liver tissues for all the treatment conditions (Fig. 8(i) and Appendix Fig. S10, S11)

Importantly, we have also shown the number of granuloma per tissue section, as quantified by the histopathology analysis of H&E stained mice lung sections in a blind study. The data is represented in the revised manuscript in Fig. 8 h. For easy reference, the CFU, granuloma and a representative image of mice H&E staining slides is also given in *rebuttal Fig. 4*. These results demonstrate that PRK treatment alone as well as in combination with Rif reduces the bacterial burden and lung-associated granuloma in the mice infected with *Mtb*, with no apparent cytotoxicity to the host tissues.

Rebuttal Fig. 4. (f) CFU analysis of *Mtb* from lungs of infected mice treated with PBS or PRK at 0 day, 15 day and 24 days of treatment. (g) CFU analysis of *Mtb* from lungs of infected mice treated with Rif and Rif + PRK at 0 day, 15 day and 24 days of treatment. Appropriate negative and positive controls taken as PBS and Rif treated mice, respectively. (n=6, six mice per condition, for all the time points and dosages and **p< 0.01 is significant). (h) Histopathology based granuloma analysis (blind) of mice lung tissues was done and the number of granulomas per tissue section was plotted for PBS treated, PRK, Rif and Rif + PRK treated mice. (i) Histopathology based H&E staining of the lung tissue sections of mice treated with PBS vs PRK, Rif and Rif +PRK treatments (representative images, detailed images in Appendix Fig. S10, S11).

4) There is no mentioning of how the authors prevented drug carry over in the mouse and macrophage cell line experiments. This is an important point, and the authors need to ensure that killing occurred inside cells/in vivo and not by residual drug on the agar plates.

For macrophage experiments, the cells were thoroughly washed with PBS post treatment and *Mtb* isolated from the infected macrophages were also washed twice.

Therefore, the chances of any external drugs remaining while plating was minimal. Also, in the Fig. 6 e,f we show through FACS analysis using a GFP labelled *Mtb*, that PRK treatment leads to a steady decline in the bacterial burden within the live macrophages. This shows that the bacterial killing is occurring inside the macrophages, and not on the plate due to drug carry over. For easy reference, the Fig. 6 e, f is also shown in rebuttal Fig. 5

Rebuttal Fig. 5. (e, f) Flow cytometry analysis of macrophage internalized *Mtb* H37Rv-GFP at 4, 8, 16 and 32 hrs post-treatment with PRK (5 μ M).

For mice experiments, the lungs were washed with PBS, homogenized and diluted several folds (1000 folds) before plating so the effective concentration of the drug, if any, would be insignificant to have any effect while plating.

Also, the decline in the lung associated granuloma by PRK treatment (*revised manuscript Fig. 8h, rebuttal Fig. 4h*) demonstrate that the killing is indeed happening inside the lungs. Therefore, in the mice experiments also, drug carryover is not likely to be the cause for bacterial killing.

We sincerely hope that reviewers will find these revisions satisfactory for publication.

References:

Asano K, Nakade S, SHIOMI T, NAKAJIMA T, SUZUKI Y, FUKUNAGA K, OGUMA T, SAYAMA K, FUJITA H, TANIGAWARA Y & ISHIZAKA A (2009) Impact of pharmacokinetics and pharmacogenetics on the efficacy of pranlukast in Japanese asthmatics. *Respirology* **14**: 822–827

Behar SM, Divangahi M & Remold HG (2010) Evasion of innate immunity by *Mycobacterium tuberculosis*: is death an exit strategy? *Nature Reviews Microbiology* **7**: 845

Beug ST, Tang VA, LaCasse EC, Cheung HH, Beaugard CE, Brun J, Nuyens JP, Earl N, St-Jean M, Holbrook J, Dastidar H, Mahoney DJ, Ilkow C, Le Boeuf F, Bell JC & Korneluk RG (2014) Smac mimetics and innate immune stimuli synergize to promote tumor death. *Nature biotechnology* **32**: 182–190

Conway GE, Casey A, Milosavljevic V, Liu Y, Howe O, Cullen PJ & Curtin JF (2016) Non-thermal atmospheric plasma induces ROS-independent cell death in U373MG glioma cells and augments the cytotoxicity of temozolomide. *British Journal of Cancer* **114**: 435–443

- Divangahi M, Desjardins D, Nunes-Alves C, Remold HG & Behar SM (2010) Eicosanoid pathways regulate adaptive immunity to Mycobacterium tuberculosis. *Nature Immunology* **11**: 751–758
- DRAZEN JM (1998) Leukotrienes as Mediators of Airway Obstruction. *American Journal of Respiratory and Critical Care Medicine* **158**: S193–S200
- Ichiyama T, Hasegawa M, Hashimoto K, Matsushige T, Hirano R & Furukawa S (2009) Cysteinyl Leukotrienes Induce Macrophage Inflammatory Protein-1 in Human Monocytes/Macrophages. *International Archives of Allergy and Immunology* **148**: 147–153
- Ichiyama T, Hasegawa M, Ueno Y, Makata H, Matsubara T & Furukawa S (2005) Cysteinyl leukotrienes induce monocyte chemoattractant protein 1 in human monocytes/macrophages. *Clinical & Experimental Allergy* **35**: 1214–1219
- Marc F, Weigel P, Legrain C, Glansdorff N & Sakanyan V (2001) An Invariant Threonine Is Involved in Self-catalyzed Cleavage of the Precursor Protein for Ornithine Acetyltransferase. *Journal of Biological Chemistry* **276**: 25404–25410
- Mayer-Barber KD, Andrade BB, Oland SD, Amaral EP, Barber DL, Gonzales J, Derrick SC, Shi R, Kumar NP, Wei W, Yuan X, Zhang G, Cai Y, Babu S, Catalfamo M, Salazar AM, Via LE, Barry CE III & Sher A (2014) Host-directed therapy of tuberculosis based on interleukin-1 and type I interferon crosstalk. *Nature* **511**: 99–103
- O'Brien J, Wilson I, Orton T & Pognan F (2003) Investigation of the Alamar Blue (resazurin) fluorescent dye for the assessment of mammalian cell cytotoxicity. *European Journal of Biochemistry* **267**: 5421–5426
- Roit FD, Engelberts PJ, Taylor RP, Breij ECW, Gritti G, Rambaldi A, Introna M, Parren PWHI, Beurskens FJ & Golay J (2014) Ibrutinib interferes with the cell-mediated anti-tumor activities of therapeutic CD20 antibodies: implications for combination therapy. *Haematologica* **100**: 77–86
- Simpson KJ, Selfors LM, Bui J, Reynolds A, Leake D, Khvorova A & Brugge JS (2008) Identification of genes that regulate epithelial cell migration using an siRNA screening approach. *Nature Cell Biology* **10**: 1027–1038
- Theron AJ, Steel HC, Tintinger GR, Gravett CM, Anderson R & Feldman C (2014) Cysteinyl leukotriene receptor-1 antagonists as modulators of innate immune cell function. *Journal of Immunology Research* **2014**: 608930–16
- Ye X-L, Lu L-Q, Li W, Lou Q, Guo H-G & Shi Q-J (2017) Oral administration of ampelopsin protects against acute brain injury in rats following focal cerebral ischemia. *Experimental and Therapeutic Medicine* **13**: 1725–1734

3rd Editorial Decision

10 January 2018

Thank you for the submission of your revised manuscript to EMBO Molecular Medicine. We have now received the enclosed reports from the referees that were asked to re-assess it. As you will see, while referee 1 is now satisfied with the revised article, referee 2 remains sceptical in parts. We would like you to carefully reply to this referee, in writing, in a point-by-point letter; we will not request additional experiments, as we agree with referee 1 that the revision is compelling enough to provide the sort of translational and novel insights we are looking for.

***** Reviewer's comments *****

Referee #1 (Remarks for Author):

Nice job of addressing the issues, I believe the authors have established that PRK is acting on target and has an interaction with the leukotriene pathway as suspected. These findings are of broad interest and are important.

Referee #2 (Comments on Novelty/Model System for Author):

It would help if the authors constructed a ArgJ mutant which should be feasible in arginine containing media and determine if this mutant is resistant against PRK activity.

Referee #2 (Remarks for Author):

In their revised manuscript Mishra and colleagues addressed many of the concerns raised during the previous review. However, the main concern about on-target activity of PRK remains. The authors demonstrate that the in vitro activity of PRK is significantly impaired by the addition of arginine supporting that PRK functions via ArgJ inhibition in vitro, however, survival was not completely restored. This observation, together with the discrepancy in activity of PRK against the purified enzyme and against whole cells indicates that PRK has one or several other targets in Mtb. Construction of an ArgJ deletion mutant (in arginine containing media) and determining the activity of PRK against the mutant would address this concern and could help in identifying additional target(s).

Moreover, the target(s) and mechanism of PRK activity against intracellular Mtb and during mouse infection remain unclear. The authors show that transcriptional regulation of genes in the 5-lipoxygenase pathway is altered by PRK, but if this contributes to the activity against Mtb is not investigated and remains unclear. Additional work is required to dissect how PRK kills Mtb in vitro, in macrophages and during mouse infection.

The in vivo CFU data reveal that during mouse infection PRK has bacteriostatic activity and only becomes bacteriocidal in combination with rifampicin. This is different from its activity against Mtb in vitro and inside macrophage-like cells. This could be dependent on the dose and a more thorough analysis of PRK's impact (dose response in vitro, in macrophages and in vivo) seems warranted.

In rebuttal Figure 3 the authors demonstrate that PRK kills Mtb in 24 hrs (~ 0.5 log reduction in CFU). It is unclear why the CFU of the untreated bacteria increase. Given that Mtb's doubling time is only ~ 22 hrs, the apparent growth (y-axis is log scale) is surprising.

The question if PRK truly binds to the allosteric pocket of ArgJ is not fully resolved. That mutation of this site leads to loss of enzymatic activity does not prove that PRK binds to that site.

The drug carry-over issue remains, washing does not necessarily remove the drugs (in fact it bears the risk of losing bacteria), which may be bound to the lipid rich mycobacterial envelope. The use of charcoal in agar plates helps to inactivate compounds that are associated with the bacteria.

3rd Revision - authors' response

19 January 2018

(begins on next page)

Comments to the reviewer

Referee #2 (Remarks for Author):

In their revised manuscript Mishra and colleagues addressed many of the concerns raised during the previous review. However, the main concern about on-target activity of PRK remains. The authors demonstrate that the in vitro activity of PRK is significantly impaired by the addition of arginine supporting that PRK functions via ArgJ inhibition in vitro, however, survival was not completely restored. This observation, together with the discrepancy in activity of PRK against the purified enzyme and against whole cells indicates that PRK has one or several other targets in Mtb. Construction of an ArgJ deletion mutant (in arginine containing media) and determining the activity of PRK against the mutant would address this concern and could help in identifying additional target(s).

We are thankful to the reviewer for appreciating the revised version.

We agree that the effect of PRK was not completely restored upon arginine supplementation. However, the efficiency of arginine uptake by *Mtb* cells from the supplemented media won't be as potent as its *de novo* synthesis by the bacterium itself. To completely restore the survival, the intra-cellular arginine levels should be same as in wildtype, which is practically difficult to reach with external supplementation. Nonetheless, by performing supplementation assay we could clearly demonstrate arginine dependent action of PRK.

Regarding the discrepancy in PRK's effect on enzyme and on whole cells, it's important to note that *in vitro* studies calculate the 'kinetic parameters' based on maximum inhibition in product formation. However, MIC/IC₅₀ are measure of inhibition induced 'cell death'. If there is an intracellular threshold of arginine required for *Mtb* survival, even slight decrease below that may lead to cell death. In such a scenario, in vivo cell death would be more prominent than *in vitro* enzyme inhibition kinetics.

Without a doubt ArgJ deletion mutant is a definitive experiment to show PRK's dependency on ArgJ. However, we are afraid that since ArgJ is essential for the survival of *Mtb* such a mutant will be lethal.

Moreover, the target(s) and mechanism of PRK activity against intracellular Mtb and during mouse infection remain unclear. The authors show that transcriptional regulation of genes in the 5-lipoxygenase pathway is altered by PRK, but if this contributes to the activity against Mtb is not investigated and remains unclear. Additional work is required to dissect how PRK kills Mtb in vitro, in macrophages and during mouse infection.

5-Lipoxygenase (5-LO) pathway is activated by *Mtb* infection and is shown to facilitate *Mtb*'s survival and pathogenesis within the host (Divangahi *et al.* 2010). We show that PRK treatment significantly downregulates host 5-LO signaling with subsequent reduction in pathogen survival. This suggests that the enhanced efficacy of PRK in infected macrophages

is *via* 5-LO signaling/leukotriene pathway. However, dissecting out the effect of PRK on these signalling pathways is tempting and will be the next best course of our studies in future.

The in vivo CFU data reveal that during mouse infection PRK has bacteriostatic activity and only becomes bacteriocidal in combination with rifampicin. This is different from its activity against Mtb in vitro and inside macrophage-like cells. This could be dependent on the dose and a more thorough analysis of PRK's impact (dose response in vitro, in macrophages and in vivo) seems warranted.

Data presented in this manuscript indicates increase in *Mtb* growth in control groups (PBS), static growth in PRK and reduced when treated with PRK + Rifampicin. However, the apparent bacteriostatic effect of PRK could be due to a steady state equilibrium between bacterial death and its division, giving an impression of bacteriostatic effect on the overall population. In combination (PRK+Rif) treatment, bacterial death rate is higher than the growth rate, hence more pronounced effect on the population. We agree with the reviewer that increasing the dose of PRK will shift the equilibrium towards higher cell death. Here we would like to emphasize that this study shows a pre-clinical, proof of principle for successful repurposing of an approved drug, for new ailment and presents a novel combination with the standard of care therapy drugs for tuberculosis. Our next line of studies will be to determine the precise dosage of PRK to have optimum effect with minimum side effects, in the *Mtb* infected mice and Guinea pigs.

In rebuttal Figure 3 the authors demonstrate that PRK kills Mtb in 24 hrs (~ 0.5 log reduction in CFU). It is unclear why the CFU of the untreated bacteria increase. Given that Mtb's doubling time is only ~ 22 hrs, the apparent growth (y-axis is log scale) is surprising.

We apologies for this confusion. The time line showed in the graph represents onset of the (PRK) treatment. Cells were incubated 48 hours before starting the treatment so day 1 for treatment is technically day three for the *Mtb* culture. We thank the reviewer for stating this confusion and have now explained this in the figure legend (Fig. 5 f,g) and in the results section.

The question if PRK truly binds to the allosteric pocket of ArgJ is not fully resolved. That mutation of this site leads to loss of enzymatic activity does not prove that PRK binds to that site.

To address this issue, we have shown the following line of evidences: we performed blind docking with the whole protein and PRK/SRB, and the inhibitor repeatedly docked into this allosteric site, in all the conformations. Next, the dye displacement assay (manuscript fig. 3 g,h) demonstrates that both PRK and SRB could displace a hydrophobic dye bound to the

allosteric pocket, in a dose dependent manner. Moreover, MD simulations showed potential residues in this pocket showing strong interaction with PRK/SRB. In the light of all these evidences we adduce that PRK binds to the mentioned pocket and inhibits *MtArgJ* allosterically.

The drug carry-over issue remains, washing does not necessarily remove the drugs (in fact it bears the risk of losing bacteria), which may be bound to the lipid rich mycobacterial envelope. The use of charcoal in agar plates helps to inactivate compounds that are associated with the bacteria.

We have tried to address the drug carryover issue by careful washing with PBS, a method generally used in the laboratories. Yet we agree that using charcoal in agar plates would have further minimized the carryover effect and we are grateful to the reviewer for this suggestion, for our future experiments.

Moreover, the concentration of PRK used in our mice model of infections is completely soluble in water (PBS) therefore we expect negligible precipitation or sticking elsewhere, in our most crucial experiments.

Comments to the editor

1) Please address the comments of referee 2 in writing in a point-by-point letter, INCLUDING my comments (in a Word file).

We have addressed all the concerns of the reviewer two in a point wise rebuttal attached along with.

2) Appendix:

please relabel Appendix Table 1->6 as Appendix Table S1 S6 and update all call outs in M&M, provide sequence of primers used for qPCR

The same has been done in this revised version.

We have provided the q-PCR primer sequences in the M&M in this revised version.

3) Legends: in figure 2 legend, please add the PDB iD 3IT6 as a reference to MtArgJ

We have added PDB id in the legend of fig. 2 this revised version

4) In the main manuscript file, please do the following:

-limit the number of keywords to 5

Done

-always indicate in legends exact n= and exact p= values, not a range, along with the statistical test used. Please make sure to populate the statistical paragraph according to all the questions asked in the author checklist that you have filled.

Done

-in M&M, for animal work, confirm that all experiments were performed in accordance with relevant guidelines and regulations. The manuscript must include a statement identifying the institutional and/or licensing committee approving the experiments. Also indicate the age, gender, starting and genetic background of animals used, along with housing conditions and methods used for culling.

Done

5) Every published paper now includes a 'Synopsis' to further enhance discoverability.

Synopses are displayed on the journal webpage and are freely accessible to all readers. They include a short stand first (maximum of 300 characters, including space) as well as 2-5 one sentence bullet points that summarise the paper. Please write the bullet points to summarise the key NEW findings. They should be designed to be complementary to the abstract - i.e. not repeat the same text. We encourage inclusion of key acronyms and quantitative information (maximum of 30 words / bullet point). Please use the passive voice. Please attach these in a separate file or send them by email, we will incorporate them accordingly.

We have attached a synopsis of our work separately with this version and will also e-mail the same.

You are also welcome to suggest a striking image or visual abstract to illustrate your article.
If you do please provide a jpeg file 550 px-wide x 400-px high.

We have also attached a visual abstract for better illustration of the work carried out in this study, separately with this version and will also e-mail the same.

Corresponding Author Name: Avadheshia Surolia

Manuscript Number: EMM-2017-08038-V3